# Arid1a restrains Kras-dependent changes in acinar cell identity

Geulah Livshits[1], Direna Alonso-Curbelo[1], John P Morris IV[1], Richard Koche[2], Michael Saborowski[3], John Erby Wilkinson[4], Scott W Lowe[1,5]*

[1]Department of Cancer Biology and Genetics, Memorial Sloan Kettering Cancer Center, New York, United States; [2]Center of Epigenetics Research, Memorial Sloan Kettering Cancer Center, New York, United States; [3]Department of Gastroenterology, Hepatology, and Endocrinology, Hannover Medical School, Hannover, Germany; [4]Department of Pathology, University of Michigan School of Medicine, Ann Arbor, United States; [5]Howard Hughes Medical Institute, New York, United States

**Abstract** Mutations in members of the SWI/SNF chromatin remodeling family are common events in cancer, but the mechanisms whereby disruption of SWI/SNF components alters tumorigenesis remain poorly understood. To model the effect of loss of function mutations in the SWI/SNF subunit Arid1a in pancreatic ductal adenocarcinoma (PDAC) initiation, we directed shRNA triggered, inducible and reversible suppression of Arid1a to the mouse pancreas in the setting of oncogenic Kras$^{G12D}$. Arid1a cooperates with Kras in the adult pancreas as postnatal silencing of Arid1a following sustained Kras$^{G12D}$ expression induces rapid and irreversible reprogramming of acinar cells into mucinous PDAC precursor lesions. In contrast, Arid1a silencing during embryogenesis, concurrent with Kras$^{G12D}$ activation, leads to retention of acinar cell fate. Together, our results demonstrate Arid1a as a critical modulator of Kras-dependent changes in acinar cell identity, and underscore an unanticipated influence of timing and genetic context on the effects of SWI/SNF complex alterations in epithelial tumorigenesis.

DOI: https://doi.org/10.7554/eLife.35216.001

*For correspondence: lowes@mskcc.org

Competing interests: The authors declare that no competing interests exist.

## Introduction

Aberrant chromatin regulation is a hallmark of cancer (*Flavahan et al., 2017*). Numerous studies have revealed an abnormal chromatin state in tumor cells in comparison with their tissue of origin, and mutations in genes encoding chromatin readers, writers and remodelers have been detected across the spectrum of human cancers (*Feinberg et al., 2016*). For example, subunits of the SWI/SNF ATP-dependent chromatin-remodeling complex show frequent mutation in many solid tumors (*Bailey et al., 2016*; *Waddell et al., 2015*; *Witkiewicz et al., 2015*), but understanding of the underlying biology and molecular mechanisms as to how these mutations influence cancer related processes remains limited.

SWI/SNF complexes regulate chromatin accessibility and thus gene expression by re-positioning nucleosomes and interacting with available transcription factors (TFs) at enhancer regions (*Kelso et al., 2017*; *Mathur et al., 2017*; *Vierbuchen et al., 2017*; *Wang et al., 2017*). While the predominance of apparently inactivating mutations in various SWI/SNF components implies a tumor-suppressive function for the complex, gene perturbation studies in mice have identified both pro- and anti-tumorigenic roles for individual SWI/SNF subunits. For instance, mice lacking Arid1a, the DNA-binding SWI/SNF subunit, in the colon develop adenocarcinomas (*Mathur et al., 2017*). However, opposing and stage dependent roles have been described within the same tissue, as Arid1a deletion can both impair and promote the initiation of liver tumorigenesis (*Fang et al., 2015*;

**eLife digest** The pancreas produces many different hormones, as well as several substances important for digestion. To perform these roles, the pancreas contains different types of cells; for example, acinar cells make digestive enzymes that help to break down food. But, like other cells in the body, pancreatic cells can accumulate mutations in their DNA that cause them to divide, acquire an altered identity and form a cancerous tumor.

The DNA of cells is packed into a structure called chromatin. While the DNA sequence is essentially the same across all normal cells of a given individual, chromatin can be more or less compacted in the different cell types that comprise our body tissues. A collection of proteins called the SWI/SNF complex can reorganize the chromatin to change how tightly the DNA is packed. This determines which genes in the DNA are accessible and can be activated, and which ones cannot.

Around 25% of pancreatic cancers contain mutations in genes that produce proteins of the SWI/SNF complex. These mutations normally occur with an additional mutation that over-activates the gene that produces a potentially cancer-causing protein called Kras. Livshits et al. have now genetically engineered mice to investigate how one such SWI/SNF complex protein, called Arid1a, affects how pancreatic cancer develops using a genetic approach that made possible to temporarily halt the production of Arid1a in acinar cells by feeding these mice an antibiotic. The gene that produces Kras was also over-activated in the pancreases of the mice, making them more likely to develop cancer.

Within just two weeks of stopping the production of Arid1a, the acinar cells stopped producing digestive enzymes and started making other proteins that are typically found in cancerous cells, indicating that Arid1a is involved in maintaining the normal identity and activity of these cells. Restoring the ability of altered acinar cells to produce normal levels of Arid1a (by removing the mice from the antibiotic diet) did not reverse these changes. Biochemical experiments showed that acinar cells with reduced levels of Arid1a have altered chromatin. In particular, the genes that produce digestive enzymes, which are normally active in healthy pancreases, were less accessible in mice who had over-active Kras and reduced levels of Arid1a.

The results presented by Livshits et al. provide the first evidence of how alterations to Arid1a can lead to irreversible changes in the identity and activity of pancreatic acinar cells. These results will need to be carefully considered by researchers who are developing treatments for cancer patients with mutations in Arid1a and other SWI/SNF proteins. In particular, methods that attempt to restore the functions of absent SWI/SNF proteins to cancer cells are unlikely to treat the cancer successfully.

DOI: https://doi.org/10.7554/eLife.35216.002

*Sun et al., 2017*) and accelerates progression and metastasis in established disease (*Sun et al., 2017*). Another subunit, Smarca4, displays the opposite pattern in pancreatic tumorigenesis, where its deletion accelerates tumor onset but slows progression (*Roy et al., 2015*; *von Figura et al., 2014a*). Thus, the role of various SWI/SNF complex components in tumor initiation and progression can be context dependent.

As the SWI/SNF complex dictates chromatin accessibility by available TFs, such context dependencies might broadly relate to development or pathological cell state. Indeed, in drosophila and mammalian systems, SWI/SNF complex dependent regulation of gene expression controls cell identity and differentiation in developmental as well as regenerative contexts (*Sun et al., 2016*; *Takada et al., 2016*; *Vieira et al., 2017*). How loss of SWI/SNF subunit function contributes to changes in cell plasticity in concert with oncogenes that drive cancer development is poorly understood. Furthermore, it is not known whether the disruption of SWI/SNF activity that accompanies tumorigenesis is needed to sustain neoplastic disease.

We set out to develop an in vivo model to address the role of the SWI/SNF component ARID1A in pancreatic ductal adenocarcinoma (PDAC), a setting in which mutations in *ARID1A* or other SWI/SNF components occur in up to 25% of cancer patients (*Shain et al., 2012*). PDAC is nearly invariably initiated by activating mutations in the *KRAS* oncogene (*Bailey et al., 2016*), while additional mutations in tumor suppressor genes are accumulated in the course of PDAC progression (*Hezel et al., 2006*). PDAC can arise from mucinous precursor lesions, including the most common,

pancreatic intra-epithelial neoplasia (PanIN), as well as intraductal papillary mucinous neoplasms (IPMN) and Mucinous Cystic Neoplasms (MCN), with activating *KRAS* mutations frequently found in these early neoplastic stages (*Hosoda et al., 2017*; *Lee et al., 2016*).

Tissue specific expression of mutant Kras in the developing and adult mouse pancreas recapitulates both the range of preneoplastic lesions and their progression to malignant PDAC (*Hingorani et al., 2005*; *Izeradjene et al., 2007*; *Sano et al., 2014*; *Siveke et al., 2007*). Lineage tracing studies indicate mutant Kras can drive PanIN development from acinar cells that undergo a process of persistent trans-differentiation termed acinar to ductal metaplasia (ADM) (*Kopp et al., 2012*). In this process, acinar cells lose their pyramidal morphology, downregulate expression of digestive enzymes and TFs characteristic of acinar cells, and turn on an embryonic progenitor-like transcriptional program that includes expression of ductal markers and development of glandular morphology (*Storz, 2017*). Deletion of key transcriptional regulators of acinar cell identity and regeneration such as *Ptf1a*, *Nr5a2* and *Pdx1*, when combined with oncogenic *Kras*, accelerates ADM and PanIN development (*Flandez et al., 2014b*; *Krah et al., 2015*; *Roy et al., 2016*; *Shi et al., 2009*; *von Figura et al., 2014b*). Thus, maintenance of acinar cell identity and regenerative homeostasis provides an important barrier to tumorigenesis in PDAC.

Current mouse models of PDAC have limitations. For example, cancer sequencing studies imply that mutations can occur in biologically-relevant orders during tumorigenesis, yet traditional Cre/loxp-based mouse models necessarily recombine all conditional alleles simultaneously. Additionally, traditional gene 'knockout' models do not allow for the restoration of the endogenous protein, making it difficult to address questions of phenotypic reversibility and the requirement for sustained gene loss in disease maintenance. Addressing these issues, we implement a powerful mouse-modeling platform that enables inducible and reversible disruption of gene function to show that Arid1a depletion can modulate the early stages of pancreatic neoplasia in a context-specific manner that depends on the timing of Arid1a inactivation and the presence of oncogenic *Kras*. Remarkably, these cell state transitions facilitated by Arid1a perturbation involve reprogramming events that, in contrast to tissue regenerative responses, are irreversible.

## Results

### A mouse model for inducible and reversible Arid1a depletion

We sought to create a model with the capability for inducible and reversible Arid1a knockdown at different stages of PDAC progression. To this end, we used a previously developed mouse model of Kras$^{G12D}$-induced pancreatic tumorigenesis that integrates robust and inducible RNA interference (RNAi) technology with embryonic stem (ES) cell genetic engineering (*Saborowski et al., 2014*). This model, referred to below as KC-RIK (*Figure 1A*), incorporates a Ptf1a-Cre that activates Cre-driven alleles in the pancreas (*Hingorani et al., 2003*), LSL-Kras$^{G12D}$ (*Jackson et al., 2001*), as well as a set of alleles that allow for doxycycline-regulatable shRNA expression: CAGS-LSL-rtTA3-IRES-mKate2 (CAGS-LSL-RIK), which enables Cre-activatable expression of a reverse tetracycline transactivator (rtTA); a red fluorescent reporter (mKate2) (*Dow et al., 2014*); and a *Col1a1* homing cassette that enables insertion of a single copy of a construct into the *Col1a1* locus via recombinase-mediated cassette exchange (RMCE) (*Beard et al., 2006*). The Ptf1a-Cre allele used in this study becomes activated in multi-potent pancreas progenitors at embryonic day 9.5 and remains active in acinar but not islet and ductal cells of the pancreas (*Kawaguchi et al., 2002*). Thus in our model, Cre recombination occurs most commonly in acinar cells, but leaves some ducts and endocrine cells un-recombined, as indicated by their lack of mKate2 staining. This ES cell system enables the direct production of experimental cohorts of chimeric mice harboring multiple alleles, thereby dramatically accelerating the rate of experimentation while simultaneously reducing animal waste as byproducts of strain intercrossing (*Dow et al., 2012*; *Premsrirut et al., 2011*).

In order to dissect the role of Arid1a in pancreatic tumorigenesis and tissue homeostasis using the above system, we first generated GFP-coupled shRNAs capable of almost complete silencing Arid1a from a single genomic copy (*Figure 1B*) and subsequently cloned two into the *Col1A1* targeting cassette downstream of a tetracycline-response element (TRE) driven promoter; this configuration enables inducible and reversible expression of a downstream transgene in cells expressing the rtTA and supplied with the tetracycline analog doxycycline (dox) (*Premsrirut et al., 2011*). KC-RIK

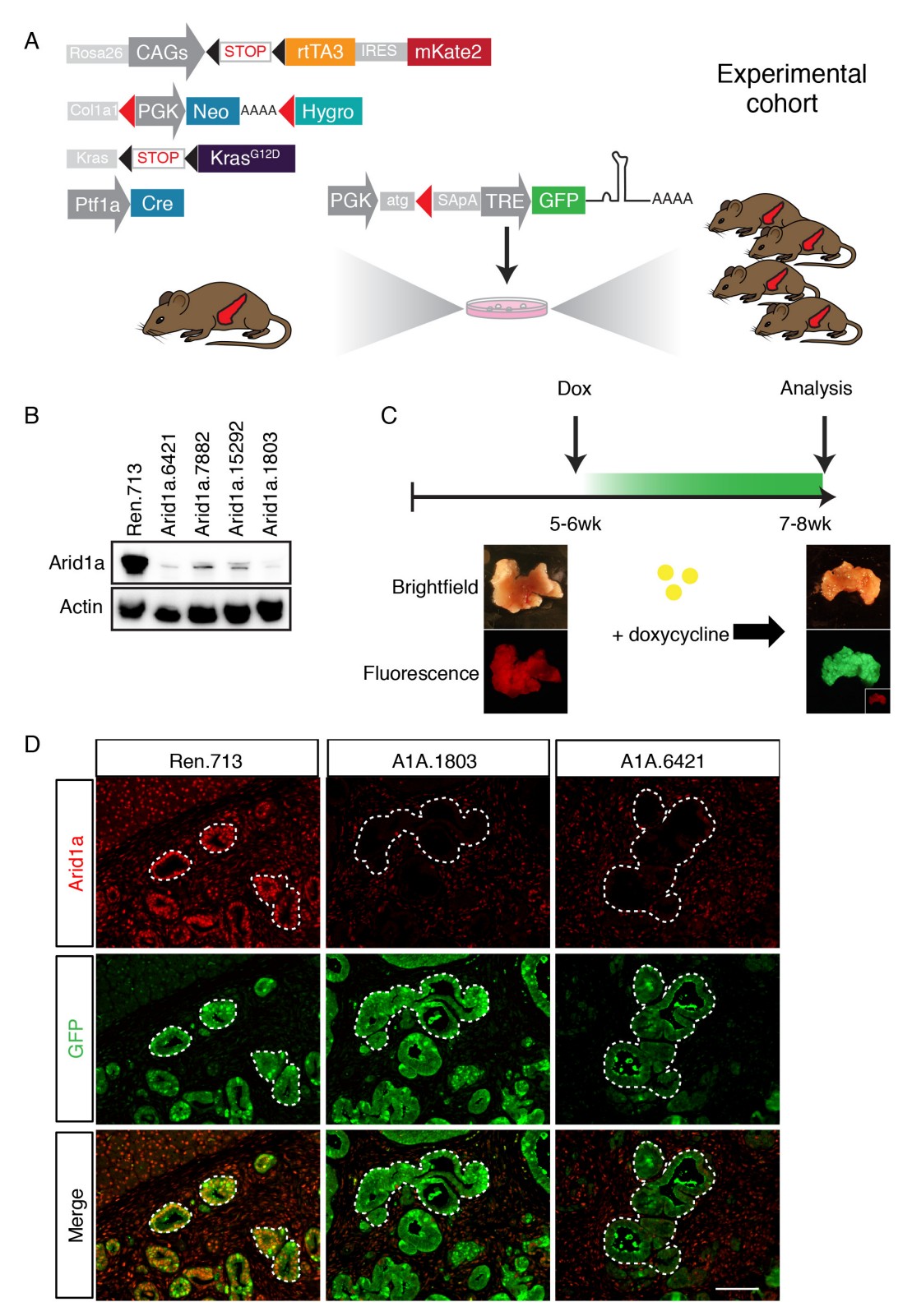

**Figure 1.** A mouse model for inducible and reversible Arid1a depletion in vivo. (**A**) Schematic of KC-RIK model. shRNAs against Arid1a and Renilla are targeted into ES cells derived from KC-RIK mice. Experimental cohorts are generated via blastocyst injection of positive clones. (**B**) Arid1a knockdown in 3T3 cells at MOI of <1. shArid1a.6421 and shArid1a.1803 were used for ES cell targeting. (**C**) Experiment schematic for acute Arid1a silencing. (**D**) GFP staining in the pancreas of KC-RIK-shRen and KC-RIK-shArid1a mice. GFP staining corresponds with shRNA expression and marks the epithelial

*Figure 1 continued on next page*

*Figure 1 continued*

compartment where p48-Cre (Ptf1a-Cre) has been expressed. Arid1a protein is detectable in GFP +regions in animals expressing the Renilla control but not Arid1a specific shRNAs. Scalebar 50 μm.

DOI: https://doi.org/10.7554/eLife.35216.003

The following figure supplement is available for figure 1:

**Figure supplement 1.** MAPK pathway activation and Arid1a depletion in KC-RIK-shArid1a mice.

DOI: https://doi.org/10.7554/eLife.35216.004

ES cells were targeted with two shRNAs against Arid1a and an shRNA against Renilla luciferase as a control. Positive clones were directly used to generate experimental cohorts of 10–20 chimeric mice per genotype via blastocyst injection. Mice ranged from 50 to 100% in ESC –derived chimerism, and animals exhibiting >90% chimerism were used for the experiments described below.

## Acute Arid1a knockdown induces rapid changes in adult acinar cell fate in the setting of Kras$^{G12D}$

We placed KC-RIK-shArid1a mice on a doxycycline (dox) diet at five weeks of age. This enabled us to suppress Arid1a in the presence of existing oncogenic *Kras* (*Bailey et al., 2016*) but *after* completion of pancreas development (*Figure 1C*, *Figure 1—figure supplement 1A,B*), mimicking somatic alteration of the SWI/SNF complex and the order of mutational events known to occur in the human disease (*Waddell et al., 2015*). KC-RIK-shRen mice were used as controls. At this age, KC-RIK-shRen pancreata exhibited a range of histological phenotypes including normal pancreatic parenchyma, acinar ductal metaplasia, a small number of early stage PanINs, and localized expansion of lesion associated fibroinflammatory stroma (*Saborowski et al., 2014*). Consistent with coupling of the Arid1a shRNA expression to a GFP reporter, we observed efficient knockdown of Arid1a in the GFP-expressing epithelial cells of the pancreas and not in the GFP- stromal cells and unrecombined duct regions (*Figure 1D*).

Surprisingly, a substantial portion of shArid1a mice became moribund within two weeks of enrollment on dox chow, while shRen mice remained healthy (*Figure 2A*). Acute morbidity was closely associated with KC-RIK-shArid1a mice exhibiting a high degree of chimerism (>95%) (*Figure 2—figure supplement 1A*), indicating that at some level normal unrecombined cells can compensate for the gross functional defects produced by Arid1a depletion in this context. Moribund mice exhibited marked steatorrhea (*Figure 2—figure supplement 1B*), indicating exocrine insufficiency as a possible cause of their failure to thrive. Consistent with decreased exocrine function, histological analyses revealed that Arid1a knockdown induces a dramatic remodeling of the exocrine pancreas compared to shRen mice. shArid1a exocrine parenchyma was predominantly replaced by Alcian Blue, Sox9, and Cytokeratin 19 positive mucinous metaplastic lesion (MMLs) (*Figure 2B–D*), characteristic of low-grade PanIN lesions (*Figure 2C*, *Figure 2—figure supplement 1G*). This increase in PanINs was accompanied by a dramatic reduction in cells positive for the differentiated acinar cell marker Cpa1 (*Figure 2E,F*), indicating a role for Arid1a in maintaining acinar cell identity in the setting of mutant Kras. Indeed, acinar cell depletion was likely a consequence of accelerated acinar ductal reprogramming rather than acinar cell death, since pancreata derived from shArid1a mice were not atrophied (*Figure 2—figure supplement 1C,D*), and showed similar levels of cleaved caspase three and Ki67 staining as controls (*Figure 2—figure supplement 1E*).

The phenotypes described above were on target, as the same changes in histology and cell fate markers were observed in mice carrying two distinct Arid1a shRNAs (*Figure 2A–F*). Furthermore, accelerated Kras driven phenotypes were observed in mice produced by standard strain intercrossing of KC-RIK-shArid1a chimeric founders, eliminating variable chimerism as responsible for the observed phenotypes (*Figure 2—figure supplement 1A*). Of note, the Alcian blue-positive PanINs induced upon Arid1a silencing maintained low levels of Arid1a even after a year on dox chow (*Figure 2—figure supplement 2A–C*). Only 1 of 11 KC-RIK-shArid1a mice that survived past 1 year showed histological evidence of PDAC at time of sacrifice (*Figure 2A*), similar to historically reported rates of PDAC development in mice harboring Kras mutations in the absence of additional genetic events (*Hingorani et al., 2003*). Thus, while the ability of Arid1a to stabilize acinar cell identity may act as a barrier to pancreatic cancer initiation, other factors are likely necessary to promote the

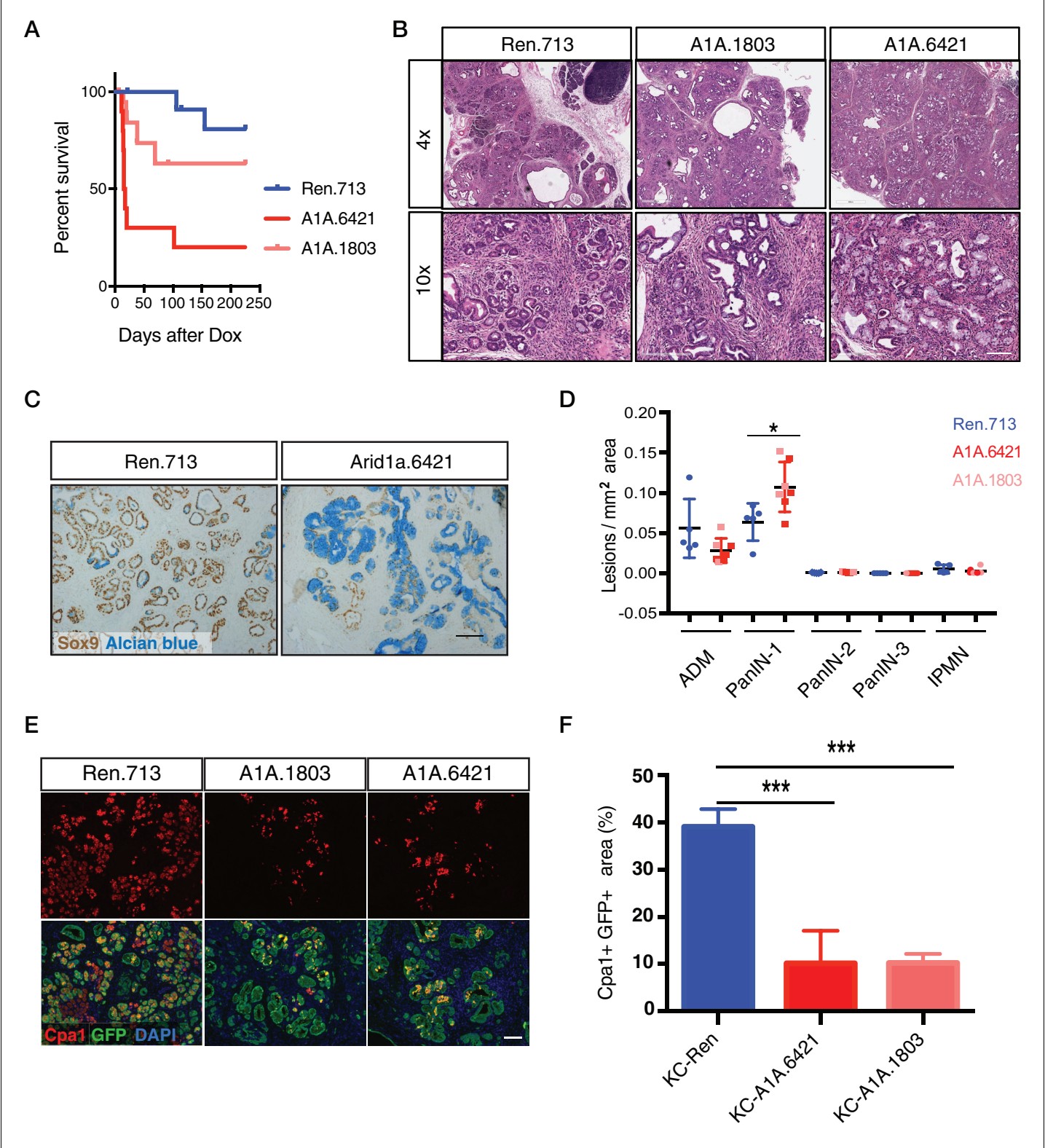

**Figure 2.** Acute Arid1a knockdown induces rapid changes in acinar cell fate in the setting of Kras$^{G12D}$. (**A**) Kaplan-Meier curve of KC-RIK-shRen and –shArid1a cohort after dox administration. (**B**) H and E staining of KC-RIK-shRen and –shArid1a mice after 2 weeks on dox. Top panels, scalebar 600 μm; bottom panels, scalebar 200 μm. (**C**) Immunohistochemistry staining for the ductal marker Sox9 in KC-RIK-shRen and –shArid1a pancreas. Alcian blue counterstain stains acidic mucins in mucinous lesions. Scalebar 100 μm. (**D**) Quantification of ADM, PanIN, and IPMN lesions per mm2 of tissue area measured by a veterinary pathologist blinded to genotype, comparing the shRen animals with the 2 Arid1a shRNA groups in the Arid1a KD genotype. *Figure 2 continued on next page*

*Figure 2 continued*

p<0.05. Each dot represents one animal. (**E**) Immunofluorescence staining KC-RIK-shRen and –shArid1a tissue for the acinar marker Carboxypeptidase 1 (Cpa1) and GFP. Nuclei are labeled with DAPI. Scalebar 100 μm.Quantification of Cpa1 staining within GFP +area in panel (**F**). p<0.0001. At least three animals per condition were used for histological quantification experiments.

DOI: https://doi.org/10.7554/eLife.35216.005

The following source data and figure supplements are available for figure 2:

**Source data 1.** Quantification of pancreatic lesions upon acute Arid1a knockdown.

DOI: https://doi.org/10.7554/eLife.35216.008

**Figure supplement 1.** Acute Arid1a knockdown impacts animal health and pancreas function in the setting of oncogenic Kras.

DOI: https://doi.org/10.7554/eLife.35216.006

**Figure supplement 2.** Histological changes induced upon acute Arid1a knockdown are maintained after 1 year on dox.

DOI: https://doi.org/10.7554/eLife.35216.007

PanIN-to-PDAC transition. Collectively, our data suggest that acute loss of Arid1a in the setting of a pre-existing *Kras* mutation rapidly perturbs the identity and function of pancreatic acinar cells, enhancing their sensitivity to metaplastic and PanIN-inducing signals stemming from oncogenic *Kras*.

## Arid1a is not required for acinar cell maintenance or regeneration in the normal pancreas

Kras driven ADM and PanIN development is constrained by programs required to maintain acinar differentiation (*Flandez et al., 2014b*; *Krah et al., 2015*; *von Figura et al., 2014b*; *Shi et al., 2009*) and regeneration (*Roy et al., 2016*). Thus, we next sought to determine whether the ability of Arid1a suppression to accelerate Kras driven preneoplastic changes in the pancreas reflects a general requirement for Arid1a function to sustain acinar homeostasis even in the absence of oncogenic signaling. To this end, we eliminated the LSL-Kras$^{G12D}$ allele from ES cell-derived KC-RIK-shArid1a and KC-RIK-shRen mice though one round of breeding. These C-RIK-shRNA mice were then placed on dox at five weeks of age, after the completion of pancreas development, as in the KC-RIK experiments. In contrast to settings involving oncogenic Kras, these mice did not become moribund following dox administration, and pancreata from both shArid1a and shRen mice appeared normal (*Figure 3A*). Histological assessment showed only subtle alterations in acinar cell morphology, with acinar rosette organization and polarity remaining intact (*Figure 3B*) despite widespread and efficient GFP expression and Arid1a knockdown throughout the pancreas (*Figure 3C,D*, *Figure 3—figure supplement 1A*). Importantly, shArid1a mice harboring wild-type Kras retained expression of the acinar marker Cpa1 (*Figure 3E,F*), which was lost in the Kras$^{G12D}$ setting. Similarly, no differences in markers of stressed acinar cells (Clusterin) or proliferation (Ki67) were noted (*Figure 3E,G,H*).

As acinar cells turn over slowly, it is possible that sufficient Arid1a protein depletion might not be achieved by short-term dox treatment. We thus maintained shRen and shArid1a mice on dox diet for 3 months. Although subtle differences in cell morphology and the pattern of E-cadherin staining were noted in cells with reduced Arid1a levels, no pancreas degeneration or exocrine insufficiency was observed despite sustained Arid1a knockdown. Therefore, Arid1a is dispensable for maintenance of normal acinar cell identiy (*Figure 3I*, *Figure 3—figure supplement 1B*).

To test if the ability of Arid1a suppression to accelerate Kras driven preneoplastic transformation is due to a defect in acinar regeneration, we challenged C-RIK-shArid1a or C-RIK-shRen mice with pancreatitis induced by the cholecystokinin analog caerulein (*Strobel et al., 2007*). Caerulein induces a reproducible cascade of tissue damage, remodeling, and repair, characterized by an inflammatory response and ADM that peak at 2 days post treatment and resolve by day 9 (*Flandez et al., 2014a*; *Greer et al., 2013*; *Morris et al., 2010*; *Shi et al., 2009*; *von Figura et al., 2014b*).

Arid1a suppression had no impact on this damage and regeneration program. At day two post-caerulein treatment, both C-RIK-shRen mice and C-RIK-shArid1a animals exhibited a similar increase in inflammation, ADM, and expression of acinar stress response genes (clusterin) and a concomitant decrease in expression of markers of acinar differentiation (Cpa1) (*Figure 3J*; and *Figure 3—figure supplement 2A–C*). Importantly, both inflammation and ADM were resolved to a similar degree by day nine in both genotypes (*Figure 3—figure supplements 2A–C* and *3A*). Regenerated acinar cells in C-RIK-shArid1a animals retained Arid1a knockdown (*Figure 3—figure supplement 3B*), ruling out the possibility that Arid1a suppressed cells were replaced by un-recombined tissue. Thus, Arid1a

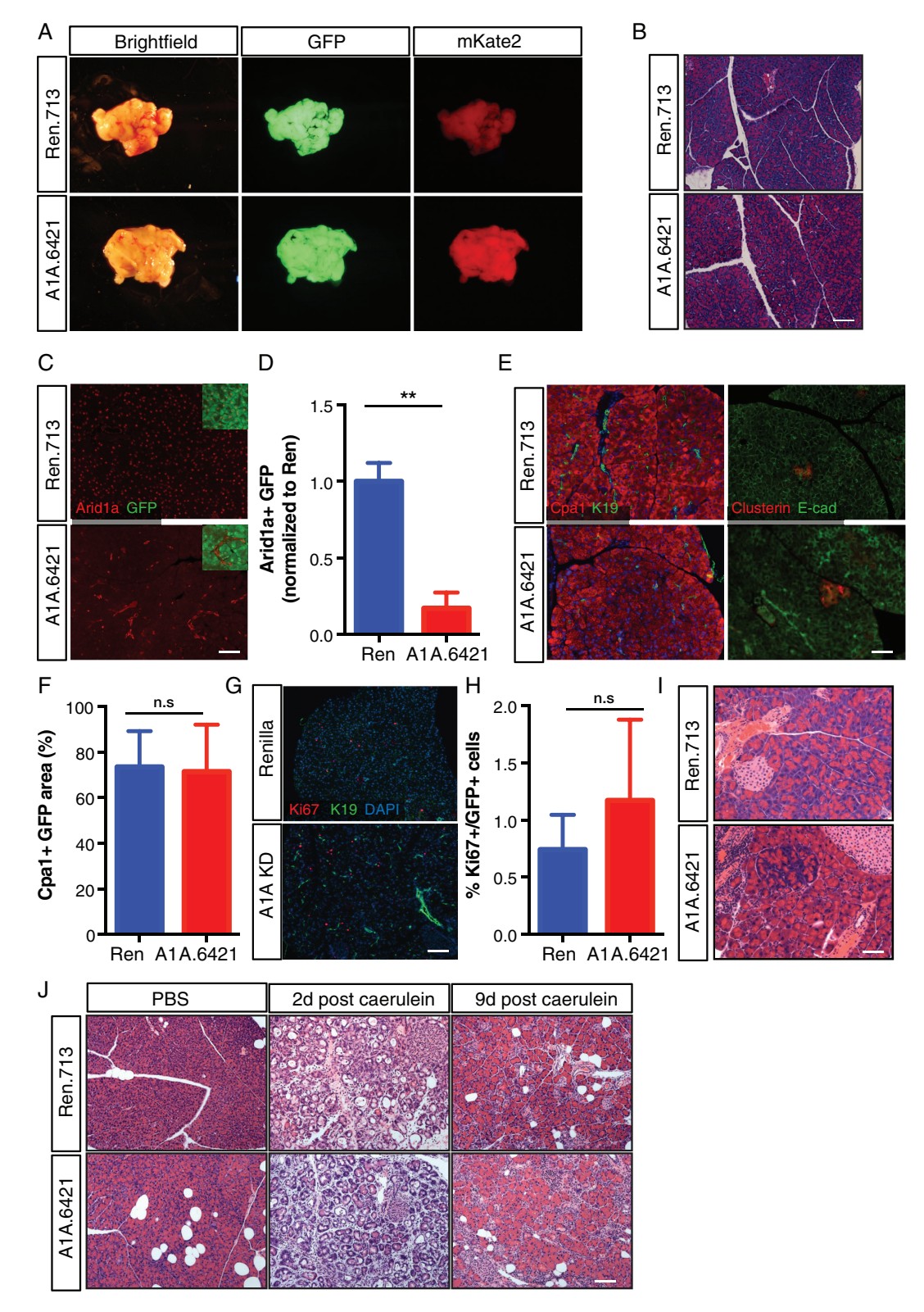

**Figure 3.** Arid1a is not required for maintenance of acinar cells in the setting of wild-type Kras. (**A**) Representative gross image of pancreas from C-RIK-shRen and C-RIK-shArid1a.6421 animals on dox for 2 weeks. (**B**) H and E staining of pancreas tissue from C-RIK-shRen and C-RIK-shArid1a.6421 animals. Scalebar 100 µm. (**C**) Immunofluorescence staining of C-RIK-shRen and –shArid1a tissue for Arid1a and GFP. Arid1a-positive cells in lower panel are GFP-negative unrecombined ducts Scalebar 100 µm. (**D**) Quantification of Arid1a + GFP + cells in panel (**C**). p<0.001. (**E**) Immunofluorescence staining

*Figure 3 continued on next page*

Figure 3 continued

of C-RIK-shRen and –shArid1a tissue. Left panel, Cpa1 and CK19. Absence of double positive cells indicates lack of ADM in shArid1a tissue. Right panel, costaining of stressed acinar cell marker Clusterin and E-cadherin. Scalebar 100 µm. (F) Quantification of Cpa1 +GFP + cells in panel (E). No significant difference. (G) Immunofluorescence staining of C-RIK-shRen and –shArid1a tissue for Ki67 and CK19. Nuclei are labeled with DAPI. Scalebar 100 µm. (H) Quantification of Ki67 +GFP + cells in panel (E). No significant difference. (I) H and E staining of pancreas tissue from C-RIK-shRen and C-RIK-shArid1a.6421 animals after 3 months on dox. Scalebar 50 µm. (J) H and E staining of pancreas tissue from C-RIK-shRen and C-RIK-shArid1a.6421 animals upon treatment with Caerulein or PBS control at indicated timepoints. Scalebar 100 µm. At least three animals per condition were used for histological quantification experiments.

DOI: https://doi.org/10.7554/eLife.35216.009

The following source data and figure supplements are available for figure 3:

**Figure supplement 1.** Arid1a depletion in the setting of wild-type Kras.

DOI: https://doi.org/10.7554/eLife.35216.010

**Figure supplement 2.** Arid1a is not required for acinar regeneration after caerulein-induced pancreatitis.

DOI: https://doi.org/10.7554/eLife.35216.011

**Figure supplement 2—source data 1.** Quantification of inflammation scores in C-RIK-shRen and C-RIK-shArid1a animals upon caerulein-induced pancreatitis.

DOI: https://doi.org/10.7554/eLife.35216.012

**Figure supplement 3.** Regeneration acinar cells in C-RIK-shArid1a mice re-express acinar markers and retain Arid1a silencing.

DOI: https://doi.org/10.7554/eLife.35216.013

function plays a genotype-specific role in dictating acinar cell identity. It is dispensable for restoring normal acinar differentiation following transient tissue damage and dedifferentiation, but plays an active role in preventing preneoplastic lineage specification driven by oncogenic Kras.

## Temporal context dictates the outcome of Arid1a depletion

Our results showing that Arid1a suppression accelerates Kras driven erosion of acinar identity are surprising in light of previous work using traditional conditional mouse models in which SWI/SNF subunits, when deleted concurrently with Kras activation, lead to decreased acinar sensitivity to Kras driven ADM and PanIN development (*Kimura et al., 2018*; *von Figura et al., 2014a*). We therefore sought to determine whether the distinct phenotypic outputs resulting from SWI/SNF subunit alterations are directly linked to their timing relative to Kras activation, or to inherent differences between experimental models. To this end, we compared the effects of suppressing Arid1a in acinar cells harboring a pre-existing Kras mutation (*Figure 2*) vs in cells in which alterations of Arid1a and Kras occur simultaneously. For the latter, we placed cohorts of chimeric shArid1a and shRen mice on dox at embryonic day 7 (E7 cohort), prior to activation of Ptf1a-driven Cre expression at day E9.5 (*Kawaguchi et al., 2002*). Thus the LSL-Kras$^{G12D}$, CAGS-LSL-RIK, and TRE-shRNA alleles are activated upon Cre expression, leading to concomitant activation of oncogenic Kras and rtTA, allowing simultaneous silencing of Arid1a in the embryos of pregnant mice treated with dox (*Figure 4A*). To compare the effects of Arid1a silencing after the onset of oncogenic Kras signaling but prior to the completion of pancreas morphogenesis, an independent cohort of mice was placed on dox at postnatal day five (P5). Other mice from this same group were left off dox to control for doxycycline effects.

In both E7 and P5 cohorts, mKate2 positive pancreas tissue was readily observed in highly chimeric on dox mice at 7 weeks of age, indicating that Arid1a silencing during pancreas morphogenesis does not in and of itself impair pancreatic cell fitness (*Figure 4B*). However, we identified marked differences at the histological level. KC-RIK-shArid1a mice from the P5 cohort showed an accelerated reduction of eosinophilic acinar cells and a concomitant increase in mucinous lesions that were morphologically consistent with low grade-PanINs, resembling the adult phenotype (*Figure 4C–E*; *Figure 4—figure supplement 1A*). In contrast, KC-RIK-shArid1a mice from the E7 cohort, while still displaying ADM/PanIN lesions, exhibited a striking retention of Cpa1$^+$ acinar tissue (*Figure 4C–E*; *Figure 4—figure supplement 1A*), a feature that was not observed in either P5 or adult cohorts. Moreover, despite efficient and comparable Arid1a knockdown in the different cohorts (*Figure 4D*, *Figure 4—figure supplement 1B*), the retained acinar cells observed in the E7 cohort failed to upregulate the ductal markers CK19 and Sox9 in the presence of mutant Kras (*Figure 4G,H*). Of note, the histology of pancreata from mice kept off dox resembled that of shRen control mice on dox,

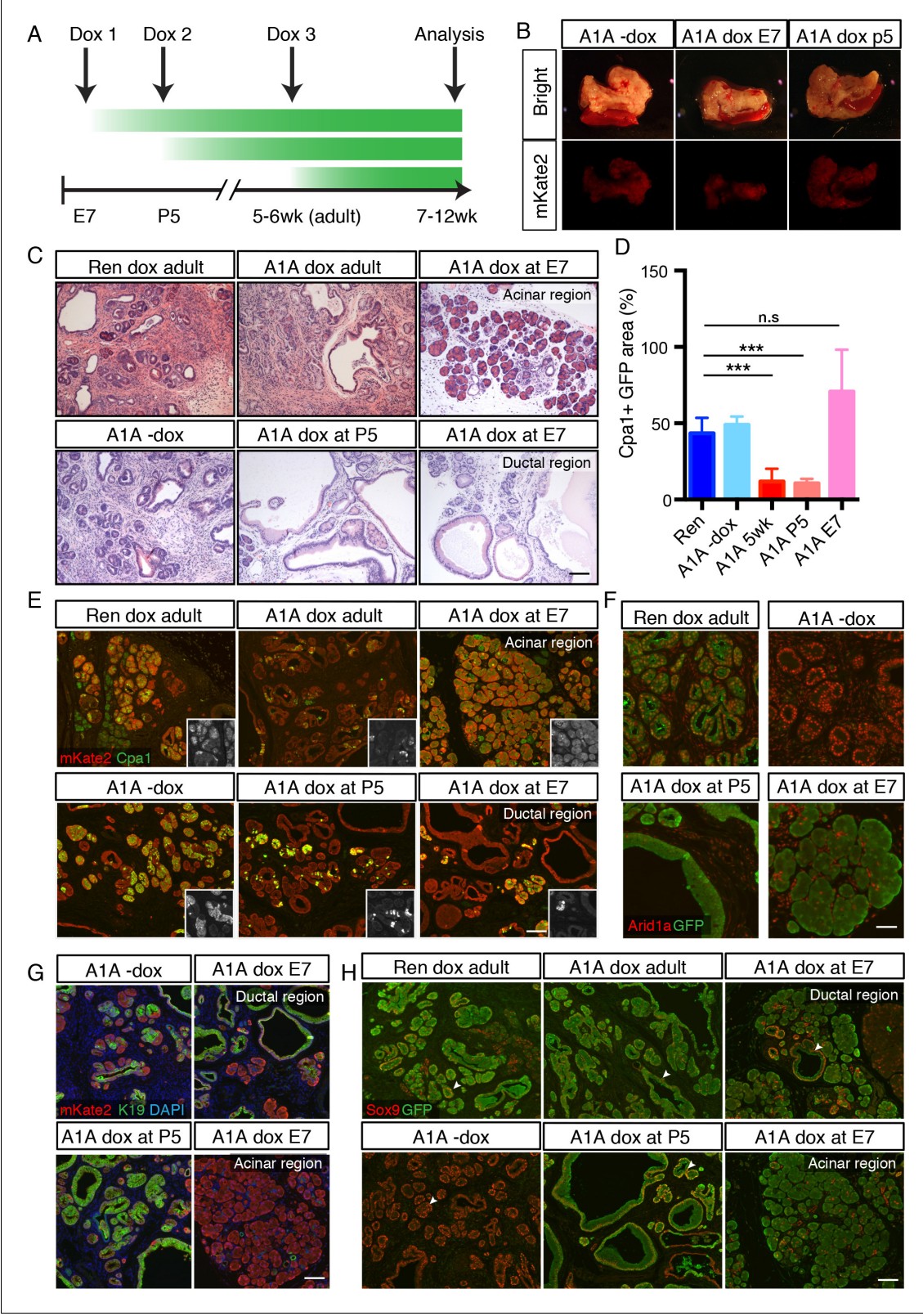

**Figure 4.** Temporal context impacts the outcome of Arid1a depletion in the pancreas. (A) Schematic of dox administration timepoints for Arid1a silencing cohorts. (B) Gross morphology and mKate2 fluorescence of KC-RIK-shRen and KC-RIK-shArid1a pancreata from indicated dox cohorts at 7 weeks of age. (C) H and E staining of KC-RIK-shRen and –shArid1a mice from indicated dox cohorts. Note the E7 dox cohort contains large mucinous lesions as well as regions of increased acinar morphology. Scalebar 100 µm. (D) Quantification of Cpa1 +GFP + cells in panel (E). ***, p<0.0001. At least

*Figure 4 continued on next page*

Figure 4 continued
three animals per condition were used for image quantification. (E) Immunofluorescence staining KC-RIK-shRen and –shArid1a tissue for the acinar marker Carboxypeptidase 1 (Cpa1) and mKate2. Insets show Cpa1 staining in grey-scale. Scalebar 100 μm. (F) Immunofluorescence staining of C-RIK-shRen and –shArid1a tissue for Arid1a and GFP. Arid1a-positive cells in lower panel are GFP-negative unrecombined ducts. Scalebar 50 μm. (G) Immunofluorescence staining KC-RIK-shRen and –shArid1a tissue for the acinar marker Carboxypeptidase 1 (Cpa1) and mKate2. Insets show Cpa1 staining in grey-scale. Scalebar 100 μm. (H) Immunofluorescence staining KC-RIK-shRen and –shArid1a tissue for CK19 and mKate2. Nuclei labeled with DAPI. Scalebar 100 μm. (I) Immunofluorescence staining KC-RIK-shRen and –shArid1a tissue for Sox9 and GFP. Arrowheads highlight GFP-positive regions with Sox9 staining in all but E7 dox areas that retain acinar morphology. Nuclei labeled with DAPI. Scalebar 100 μm.
DOI: https://doi.org/10.7554/eLife.35216.014
The following source data and figure supplements are available for figure 4:
Figure supplement 1. Mucinous lesions in KC-RIK-shArid1a mice placed on dox during embryonic development.
DOI: https://doi.org/10.7554/eLife.35216.015
Figure supplement 1—source data 1. Quantification of lesions in KC-RIK-shArid1a and control animals placed on dox during pancreatic development.
DOI: https://doi.org/10.7554/eLife.35216.016

ruling out potential effects of the dox diet itself (*Figure 4C–F*). Thus, Arid1a suppression accelerates the transition from an acinar to a mucinous cell fate in the presence of a *pre-existing* oncogenic Kras mutation (adult and P5 cohorts) whereas, when occurring *concomitant* with Kras$^{G12D}$ activation, it paradoxically promotes retention of the acinar cell phenotype (E7 cohort). Taken together, these data reveal a temporal, context-dependent role for Arid1a in regulating acinar identity in response to oncogenic Kras.

## Acinar to PanIN transition is not reversed upon Arid1a restoration

In addition to uncoupling the timing of oncogene activation and tumor suppressor gene loss, our system is reversible, and as such enables assessment of whether ongoing gene suppression is needed to sustain pathological phenotypes. To test the stability of the acinar-to-mucinous reprogramming triggered by Arid1a depletion, we placed cohorts of KC-RIK-shRen and KC-RIK-shArid1a mice on dox at 5 weeks of age for two weeks to establish the acute Arid1a phenotype. Mice were then withdrawn from doxycycline to restore Arid1a expression (*Figure 5A*).

Remarkably, Arid1a restoration did not reverse the acinar reprogramming phenotype. As expected, pancreata from mice withdrawn from dox retained expression of the mKate2 lineage marker, but lost the shRNA-linked GFP fluorescence signal (*Figure 5B*). Staining of tissues from KC-RIK-shRen and -shArid1a animals for Arid1a showed partial restoration of Arid1a by 5 days after withdrawal and full restoration at 2 and 4 weeks (*Figure 5C*). Despite this, histological assessment of KC-RIK-shRen and -shArid1a mice after dox withdrawal showed retention of widespread mucinous lesions as in the on dox condition, without a reversal of the PanIN phenotype towards an acinar or even ADM cell fate (*Figure 5D,E*). Alcian blue staining further confirmed the retention of mucinous lesions in Arid1a-restored mice (*Figure 5E*). Thus, Arid1a loss enables a *non-reversible* reprogramming of cell fate that is dependent on cellular and genetic context.

## Acute Arid1a loss results in rapid transcriptional shifts in cell identity

We next sought to understand the mechanism(s) whereby Arid1a regulates pancreas cell identity. Perturbation of the SWI/SNF chromatin-remodeling complex is expected to impact the expression of SWI/SNF target genes (*Kadoch and Crabtree, 2015*). In order to capture the immediate transcriptional consequences of Arid1a silencing, we performed gene expression analysis via RNAseq on mKate2$^+$GFP$^+$ epithelial cells sorted from 3 KC-RIK-shRen, or shArid1a and mice after 5 days of dox treatment. This timepoint occurs after maximal Arid1a suppression but before any observed histological changes, thus reducing the confounding effect of secondary changes in gene expression resulting from tissue inflammation (*Figure 6—figure supplement 1A*). Principal component analysis of RNA-seq data indicated that the two shRNAs targeting Arid1a conferred a transcriptional profile distinct from that of the shRen animals, confirming these shRNAs triggered minimal off-target effects (*Figure 6A*). Unsupervised hierarchical clustering of samples based on differentially expressed genes also segregated the two shRNAs away from shRen (*Figure 6B*).

Further analysis of the RNAseq data using an FDR threshold of 0.05 detected 711 downregulated and 505 upregulated genes (*Figure 6C*). Among the notable changes were a reduction in acinar-

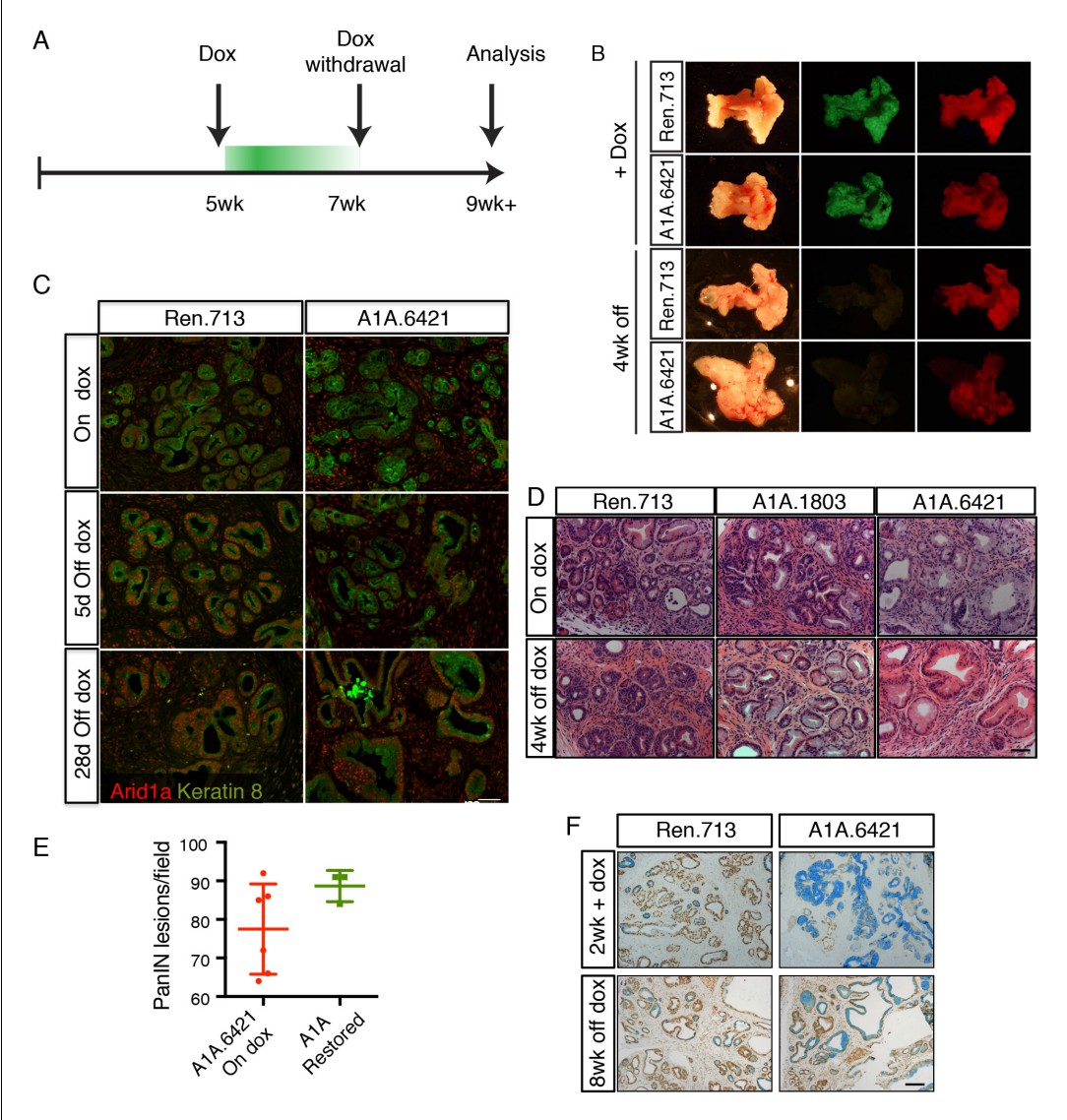

**Figure 5.** Perturbation of acinar cell fate is not reversed upon Arid1a restoration. (**A**) Schematic of dox administration timepoints for Arid1a restoration. (**B**) Gross morphology and mKate2 fluorescence of KC-RIK-shRen and KC-RIK-shArid1a pancreata on dox and after 4 weeks of dox withdrawal. (**C**) Immunofluorescence staining of C-RIK-shRen and –shArid1a tissue for Arid1a and the ductal marker Keratin 8. (**D**) H and E staining of KC-RIK-shRen and –shArid1a mice on dox and after 4 weeks of dox withdrawal. (**E**) Quantification of PanIN lesions per field in (**D**) measured by veterinary pathologist. No significant difference between KC-RIK-shArid1a (6421) mice on dox and upon dox withdrawal. (**F**) Representative immunohistochemistry staining for the ductal marker Sox9 in pancreata from KC-RIK-shRen and –shArid1a.6421 mice from same cohort quantified in (**E**) Alcian blue counterstain stains acidic mucins in mucinous lesions. Scalebars 100 μm.

DOI: https://doi.org/10.7554/eLife.35216.017

related digestive enzymes and an increase in cellular components needed for the mucin production characteristic of PDAC precursor lesions. Consistent with our histological findings, gene set enrichment analysis (GSEA) revealed a general activation of a mucinous lesion transcriptional program after 5 days on dox: genes that are decreased in PanIN organoids relative to normal organoids were also downregulated, while genes enriched in PanIN organoids were up in shArid1a pancreata as well (*Figure 6D*) (*Boj et al., 2015*). Selected genes within the upregulated set included known PanIN and PDAC markers such as Agr2 and Mesothelin, which were confirmed by RT-qPCR (*Figure 6E*). Interestingly, gene sets associated with proliferation and 'E2F' were also enriched despite no detectable increase in proliferation at the timepoints analyzed. These observations raise the possibility that

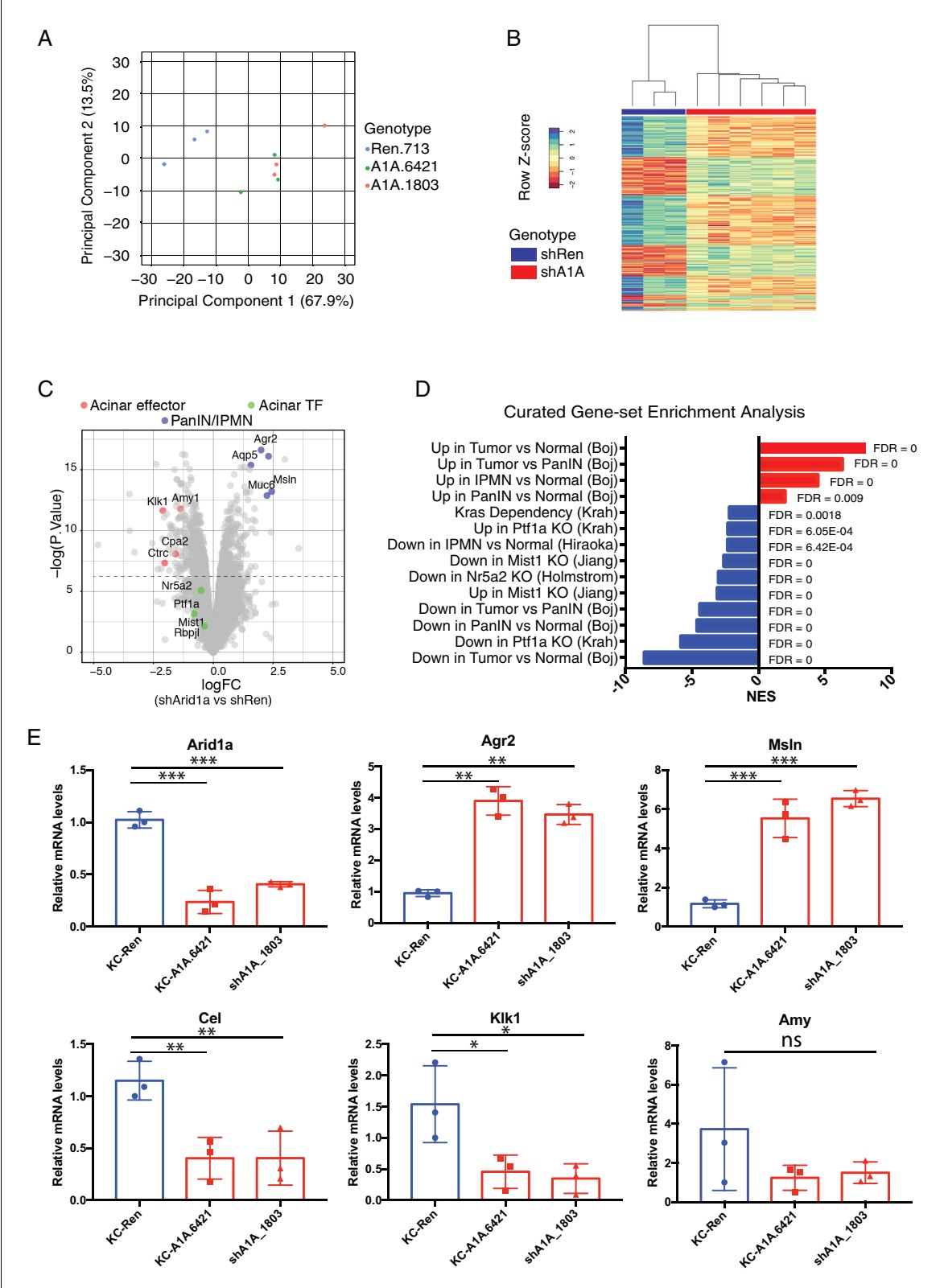

**Figure 6.** Acute Arid1a loss results in rapid transcriptional shifts in cell identity. (**A**) Principal component analysis of RNAseq data for KC-RIK-shRen, -shArid1a.6421 and –shArid1a.1803 animals after 5 days on dox. three animals per group. (**B**) Unsupervised clustering of samples by genes differentially expressed between shRen and shArid1a groups clusters the two Arid1a shRNAs together. (**C**) Volcano plot of significant differentially expressed genes upon Arid1a knockdown (FDR threshold <0.05 and log2FC cutoff of 1). (**D**) Literature-curated gene sets altered upon Arid1a knockdown; Sources listed

*Figure 6 continued on next page*

Figure 6 continued

in parentheses. (E) qPCR assessment of acinar effectors and genes associated with mucinous pancreatic lesions after 5 days of dox treatment (same timepoint as RNAseq) in shRen and shArid1a animals (n = 3 per genotype). * Shown are mean values ± standard deviation from three biological replicates (independent animals). *p<0.05, **p<0.01, ***, p<0.0001.

DOI: https://doi.org/10.7554/eLife.35216.018

The following source data and figure supplement are available for figure 6:

**Source data 1.** QPCR validation of transcriptional changes observed by RNAseq.
DOI: https://doi.org/10.7554/eLife.35216.020
**Figure supplement 1.** Transcriptional changes induced upon Arid1a loss are reflected by histological expression changes.
DOI: https://doi.org/10.7554/eLife.35216.019

Arid1a depletion in this context 'primes' acinar cells for tumorigenesis by facilitating shifts in cell identity, but additional inactivation of tumor suppressor pathways may be required for PanINs to acquire increased proliferative capacity associated with malignant transition to PDAC (*Caldwell et al., 2012*).

In parallel with mucin upregulation characteristic of PanIN development in KC-RIK-shArid1a mice, GSEA identified downregulation of genes associated with acinar function, such as those encoding the digestive enzymes Cel, Klk1 and Amy1 (*Hoang et al., 2016*), which was confirmed by RT-PCR (*Figure 6E*). Importantly, significant downregulation of gene expression programs sustained by master acinar TFs (Nr5a2 and Ptf1a) required to maintain acinar differentiation (*Holmstrom et al., 2011*; *Jiang et al., 2016*) (*Figure 6D*) were noted. Genes with altered expression at the transcript level showed parallel changes at the protein level (*Figure 6—figure supplement 1B*). Interestingly, our RNA-seq data did not show significant downregulation of the acinar TFs Nr5a2, Ptf1a or Mist1 themselves at this timepoint (see *Figure 6C*). Thus, the destablization of acinar cell identity that occurs upon Arid1a suppression in the setting of oncogenic Kras most likely results from impaired recruitment or function of these acinar TFs at their appropriate targets.

## Acute Arid1a silencing decreases chromatin accessibility

Building on recent observations (*Bossen et al., 2015*; *Mathur et al., 2017*; *Miller et al., 2017*), we hypothesized that the transcriptional changes induced by Arid1a loss may be triggered by changes in chromatin accessibility. To test this in the context of pancreas tumor initiation, we performed the Assay for Transposase-Accessible Chromatin followed by high-throughput sequencing (ATAC-seq) (*Buenrostro et al., 2013*) on mKate2$^+$GFP$^+$ cells sorted from KC-RIK-shRen and shArid1a mice after 5 days on dox, as in our transcriptional profiling experiments. ATAC-seq assesses chromatin accessibility genome-wide, and therefore provides a direct readout of the chromatin remodeling activity of SWI/SNF complexes. Importantly, ATAC-seq requires substantially fewer cells than current ChIP-seq protocols and, as such, is appropriate for analyzing chromatin states in the limited cell numbers that can be recovered from the preneoplastic murine pancreas.

We found that knockdown of Arid1a resulted in striking changes in chromatin accessibility, with shArid1a chromatin profiles that clearly separated from shRen by PCA (*Figure 7A,B*). 50,000–60,000 dynamic peaks were observed, virtually all of which were reduced in the shArid1a samples relative to the shRen controls (*Figure 7B,C*). This indicates that Arid1a-containing SWI/SNF complexes are required to maintain substantial regions of open chromatin in the pre-neoplastic pancreas. Interestingly, although the chromatin accessibility was highest around transcriptional start site (TSS) regions (*Figure 7—figure supplement 1A*), the majority of dynamic sites were located in intergenic regions or introns, while promoters displayed fewer dynamic peaks (relative to the genome-wide distribution of total detectable peaks) (*Figure 7—figure supplement 1C,D*). In B cells, intergenic and intronic regions that are enriched for Brg1 binding and lost upon Brg1 ablation are associated with H3K4me1 bound enhancers essential for establishing B-cell fate (*Bossen et al., 2015*). Accordingly, we suspect that the regions of reduced chromatin accessibility observed upon Arid1a knockdown includes enhancers, whose inactivation may be ultimately responsible for the altered transcriptional profiles we observed.

To define the TFs that might be responsible for these changes, we searched for transcription factor motifs enriched in regions genomic regions showing reduced accessibility following Arid1a suppression using the HOMER suite of tools for motif discovery (*Heinz et al., 2010*). Several motifs

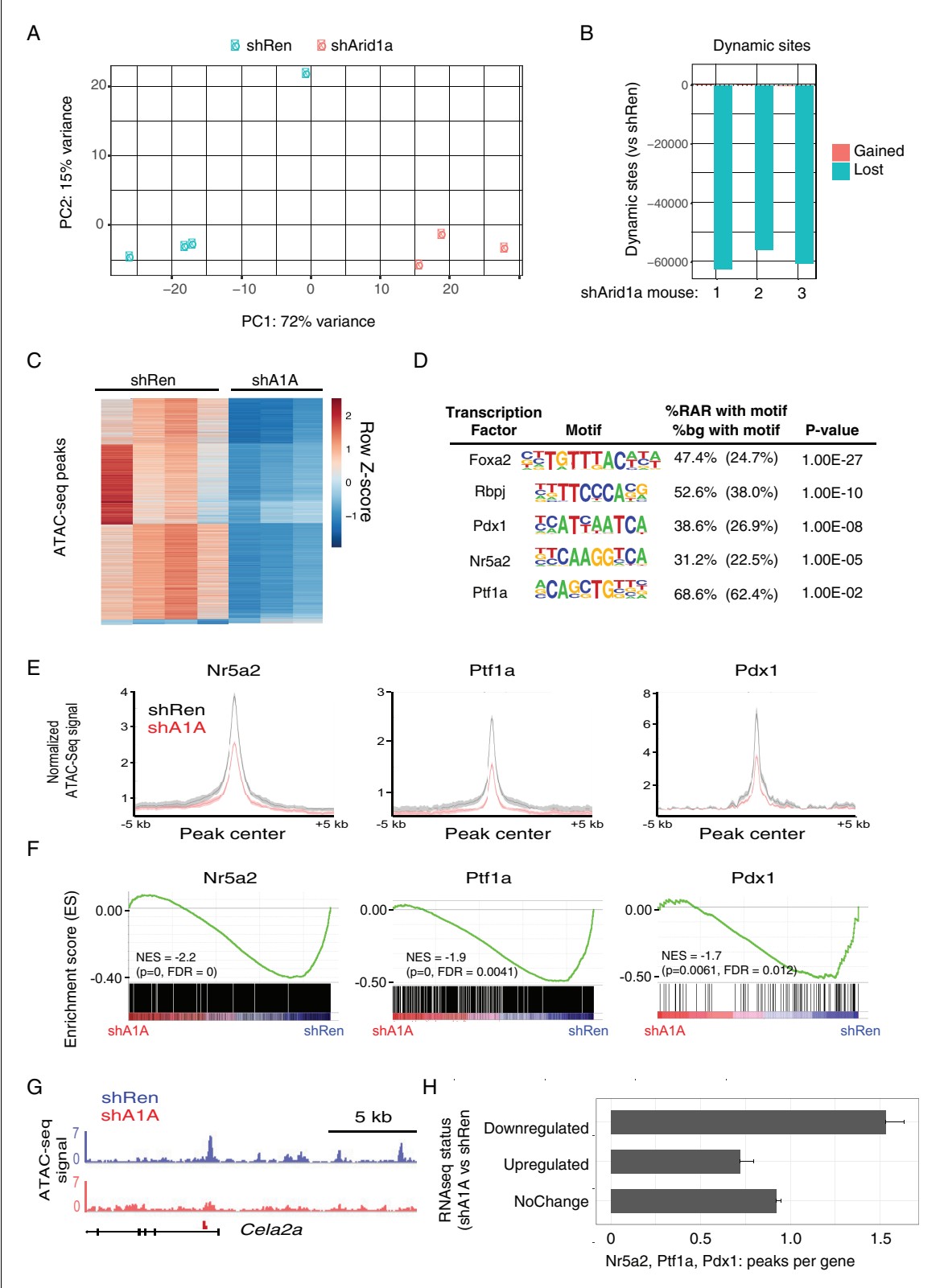

**Figure 7.** Acute Arid1a silencing decreases chromatin accessibility. (**A**) Principal component analysis of ATACseq data for KC-RIK-shRen and shArid1a.6421 animals after acute silencing. (**B**) Relative comparison of dynamic ATACseq peaks in acute KC-RIK-shArid1a.6421 animals vs one shRen control animal. Shown are the number of peaks that are gained (red) or lost (blue) in three independent shArid1a animals plotted in comparison to a representative shRen animal. Virtually all dynamic peaks are reduced in shArid1a pancreas. (**C**) Heatmap illustrating accessibility levels of dynamic peaks

*Figure 7 continued on next page*

*Figure 7 continued*

across KC-RIK-shRen and shArid1a.6421 animals. (D) Pancreatic transcription factor motifs (HOMER) enriched in regions of reduced accessibility in shArid1a. (E) ATAC-seq metaprofile of chromatin accessibility in shRen vs shArid1a cells across a 5 Kb window centered on previously defined binding sites for the indicated TFs. GSE accession numbers of ChIP-Seq datasets used are indicated in *Figure 7—figure supplement 1D*. (F) GSEA showing reduced expression of genes associated with peaks bound by the indicated TFs and losing accessibility in shArid1a.6421 vs shRen mice. (G) Example gene with reduced accessibility peak away from TSS in shArid1a pancreas. (H) Bar plot of TF density indicating the number of Ptf1a, Nr5a2, or Pdx1 peaks per gene with reduced chromatin accessibility upon Arid1a suppression. Genes were grouped by expression status in shA1A vs shRen RNA-seq as upregulated, downregulated, or stable. Error bars represent standard error of the mean.

DOI: https://doi.org/10.7554/eLife.35216.021

The following figure supplement is available for figure 7:

**Figure supplement 1.** Peaks with decreased accessibility upon Arid1a depletion are enriched for intronic/intergenic regions and acinar TF binding sites.

DOI: https://doi.org/10.7554/eLife.35216.022

showed significant enrichment in these reduced accessibility regions (RARs), including AP1/Jun motifs, which exhibit SWI/SNF dependent binding in other systems (*Figure 7—figure supplement 1C*) (*Kelso et al., 2017*; *Mathur et al., 2017*; *Sun et al., 2016*). In addition, and consistent with impact of Arid1a suppression on acinar cell fate, we identified enrichment of binding motifs for several key acinar-specific TFs (Nr5a2, Ptf1a and Rbpj) in the genomic regions losing chromatin accessibility upon Arid1a knockdown (*Figure 7D*). These same sites also showed enrichment in binding motifs for Pdx1, a TF that constrains acinar identity during mutant Kras-driven PanIN development (*Roy et al., 2016*) (*Figure 7D*).

To further connect these chromatin changes with the transcriptional repression of the acinar identity genes identified above (*Figure 6C*), we performed a comprehensive analysis of gene expression and chromatin accessibility at sites of the pancreatic TFs Nr5a2, Ptf1a and Pdx1, using high confidence, experimentally-defined datasets of TF binding sites defined by ChIP-Seq (*Holmstrom et al., 2011*; *Hoang et al., 2016*; *Roy et al., 2016*). Critically, loss of these TFs destabilizes acinar differentiation and accelerates the development of Kras driven metaplasia and PanINs (*Krah et al., 2015*; *Roy et al., 2016*; *von Figura et al., 2014b*). This analysis showed a significant enrichment of TF binding sites at ATAC-seq peaks that lose accessibility upon Arid1a knockdown compared with sites without loss of acessibility (Nr5a2 and Ptf1a: $p<2.2e-16$, Pdx1: $p<0.0016$; Fisher's exact test; see *Figure 7E* and a representative ATAC-Seq track for the acinar target *Cela2a*). Increased chromatin closing observed upon Arid1a knockdown at these TF target sites was associated with reduced gene expression at adjacent genes. As shown in *Figure 7F*, Nr5a2, Ptf1a and Pdx1 target genes losing chromatin accessibility upon Arid1a suppression were significantly downregulated in shArid1a vs. shRen mice. More globally, genes downregulated upon Arid1a suppression showed a significantly increased density of Nr5a2, Ptf1a and Pdx1 binding sites losing chromatin accessibility as compared to genes whose expression is upregulated or unchanged (*Figure 7G*; Mean peak density: downregulated genes 1.53, upregulated genes: 0.72, unchanged: 0.92; Welch's t test: $p=2.55e-14$ for down vs. upregulated genes, $p=6.372e-11$ for down vs. unchanged genes). These data support a model whereby Arid1a depletion results in loss of chromatin accessibility at acinar-specific enhancers, limiting the transcriptional output of acinar master TFs and, consequently, the expression of transcriptional programs that direct acinar cell fate.

## Discussion

Mutations in chromatin remodelers are prevalent across the human cancer spectrum, making identification of tissue specific relationships between chromatin regulation and tumorigenesis an important goal in understanding neoplastic progression. In this study, we used a rapid and robust mouse-modeling platform to study the consequences of inducible and reversible suppression of the Arid1a SWI/SNF subunit in a mouse model of pancreatic cancer. Suppression of Arid1a in adult acinar cells harboring oncogenic *Kras* mutations accelerates acinar to ductal reprogramming and specification of mucinous, pancreatic cancer precursor lesions that cannot be reversed by restoring endogenous Arid1a expression. Thus, our data suggest that Arid1a can provide a potent barrier to Kras driven acinar to ductal metaplasia and, consequently, early stage pancreatic neoplasia. Once disrupted by

Arid1a mutation, these altered cells are reprogrammed in such a manner that Arid1a is no longer needed to sustain the aberrant cell fate.

The phenotype produced by Arid1a depletion in the presence of an oncogenic *Kras* mutation is characterized by reduced chromatin accessibility accompanied by loss of acinar morphology, reduced expression of acinar genes, and upregulation of a signature characteristic of mucinous lesions. These features imply that genomic areas rendered accessible by Arid1a harbor elements involved in preventing acinar to duct like trans-differentiation in the context of mutant Kras, a critical rate-limiting step in Kras driven pancreatic neoplasia. Indeed, our transcriptional and chromatin profiling data imply that Arid1a disruption, in the presence of oncogenic Kras, reduces the chromatin accessibility of key TFs to the enhancers of genes needed to sustain acinar cell fate. At least one such factor, Nr5a2, can physically interact with the SWI/SNF complex via the Baf60 subunit (*Debril et al., 2004*), suggesting a possible mechanism for aberrant Nr5a2 target deregulation upon Arid1a loss. Further studies will be necessary to determine the interplay between Nr5a2 and other acinar TFs that may underly the Arid1a-depletion phenotype.

The accelerated PanIN formation produced by Arid1a suppression required the presence of a Kras mutation, the initiating event in pancreatic carcinogenesis. Thus, Arid1a suppression in the normal pancreas did not appreciably perturb acinar differentiation, even during wound healing responses that force a reversible acinar to ductal metaplasia. Apparently, Arid1a suppression sensitizes pancreatic epithelial cells to mucinous reprogramming by oncogenic Kras or, alternatively, increases the efficiency of Kras-induced acinar-to-ADM-to-PanIN transitions. Regardless, this Kras-dependence differs from the phenotype produced by deletion of acinar TFs such as Ptf1a or Nr5a2, whose loss generally perturbs acinar maintenance and/or regeneration (*Cobo et al., 2018*; *Flandez et al., 2014a*; *Krah et al., 2015*; *von Figura et al., 2014b*). We hypothesize that the potent and cooperative effects between oncogenic signaling and chromatin remodeling at eroding barriers to cellular plasticity contributes to the high frequency of SWI/SNF component inactivation in many cancer types.

The enhanced PanIN phenotype observed in our model has intriguing differences and commonalities with other models harboring conditional deletions of different SWI/SNF components. Deletion of Brg1 during pancreatic development in conjunction with oncogenic Kras results in the formation of IPMN-like lesions contiguous with the native duct system that may derive from the ductal compartment rather than through ADM. Furthermore elimination of Brg1 in adult acinar cells prevents the development of Kras driven acinar to duct-like transdifferentiation and PanIN development (*von Figura et al., 2014a*). While a recent study employing conditional deletion of Arid1a in the pancreas also observed IPMN formation, PanIN formation was not suppressed, pointing towards distinct molecular consequences of targeting different SWI/SNF components (*Kimura et al., 2018*). The striking predominance of PanINs over IPMNs in our model may be due to a dosage effect, where IPMN formation requires complete loss of Arid1a in ductal progenitors or ductal cells, while acinar cells expressing mutant Kras are more highly dependent on Arid1a to maintain their identity, and thus are rapidly reprogrammed to mucinous PanINs upon partial Arid1a suppression. This has important implications for human cancers, which exhibit a broad spectrum of Arid1a levels.

Unlike our Arid1a suppression strategy, mouse models using conditional alleles necessarily disrupt SWI/SNF concomitantly with Kras activation. Indeed, when we recapitulate such timing during embryonic development, we see a lower level of PanIN formation with shArid1a pancreata showing the retention of acinar cells, that contrasts with the effects of sequential suppression of Arid1a in acinar cells harboring pre-existing expression of mutant Kras. While the distinct molecular consequences of different SWI/SNF component mutations may contribute to phenotypic variability in other aspects of PDAC progression, these observations indicate that the timing of SWI/SNF perturbation can have a dramatic effect on tumorigenesis phenotypes. Whether the temporal effects of SWI/SNF action are a consequence of developmental stage or priming events produced by the prior presence of activated Kras remains to be determined. In any case, it seems likely that one or both processes influences the transcription factor milieu and, accordingly, the specific consequences of SWI/SNF disruption (*Roy et al., 2015*; *Sun et al., 2017*). Regardless of the precise mechanism, the fact that Arid1a suppression can produce distinct phenotypes depending on the timing of gene suppression has ramifications for studying the SWI/SNF complex in other cancer types and, more generally, for cancer modeling in the mouse. Indeed, human cancers evolve as a consequence of a stepwise accumulation of mutations rather than the acquisition of multiple mutations simultaneously.

Consequently, some phenotypes attributed to particular gene mutation in mice may not always extrapolate to the human system – not because of species differences, but because of the differential consequences of the timing of genetic events. Efforts to combine different recombination systems (e.g. Cre/Loxp and Flp/Frt) provide one way to uncouple events (*Akagi et al., 1997*; *Orban et al., 1992*). Additionally, our approach, which uses Cre-specific recombination to produce some oncogenic events and the tetracycline system to dictate timing of others, provides another. Such models may develop tumors with progression kinetics or therapeutic sensitivity profiles that more closely resemble those of patients.

Another unique advantage of our mouse modeling system is its ability to evaluate the reversibility of loss-of-function phenotypes, an approach that helps elucidate whether a cancer promoting event is required to sustain established disease. Indeed, the inducible shRNA technology used herein has been optimized to produce gene suppression that approaches that produced by gene deletion (*Bolden et al., 2014*; *Dow et al., 2015*; *Ebbesen et al., 2016*; *Hemann et al., 2003*; *Premsrirut et al., 2011*) yet, by leaving the target locus intact, enables the endogenous protein to be re-expressed upon shRNA silencing. Here, shRNA mediated knockdown of Arid1a produced a shift in acinar to mucinous cell fate and a PanIN histopathology that was not reversed by subsequent Arid1a re-expression. Such results suggest that ectopic overexpression studies in cell lines with mutations in SWI/SNF components may not reflect the biological activity of that component during tumor evolution. Furthermore, they stand in contrast to what has been noted for other tumor suppressor genes, for example, in colon cancer models where reversible APC suppression promoted the expansion of undifferentiated cells that readily underwent differentiation upon APC restoration (*Dow et al., 2015*). Since ARID1A controls the expression of thousands of gene targets via interaction with available TFs (*Kadoch and Crabtree, 2015*; *Kelso et al., 2017*; *Mathur et al., 2017*; *Sun et al., 2016*), it is possible that factors absent from mucinous lesions might be needed to reestablish an acinar transcriptional program when Arid1a is restored. Although Kras$^{G12D}$ itself prevents acinar regeneration after pancreatitis-induced ADM, expression of the bHLH protein E47 in PDAC cells induces expression of digestive enzymes and activates a Mist1-based transcriptional network, indicating that such a reversion is possible (*Kim et al., 2015*).

Taken together, our data identify a role for Arid1a in the early stages of PDAC. This supports a model where initiating oncogenic *KRAS* mutations trigger ADM and PanIN conversion at low efficiency; mutation of *ARID1A* may boost the frequency of this process by increasing cellular plasticity. Other tumor suppressor mutations (such as *TP53* and *CDKN2A*) that co-occur with *KRAS* and *ARID1A* can then contribute to further PanIN progression. While studies to explore the role of *ARID1A* in later stage PDAC progression are in progress, the irreversibility of the Kras-induced PanIN phenotype indicates that impaired SWI/SNF function may not be required for tumor maintenance. If confirmed, the identification of genotype-specific dependencies produced by SWI/SNF dysfunction rather than restoration of its normal activity may be a more effective way to exploit SWI/SNF mutations for new therapeutic approaches. While some proposed vulnerabilities, such as ARID1B, extend across multiple tissues and mutational landscapes (*Helming et al., 2014*), our data imply that many such dependencies are likely to be limited to a particular cell type or co-mutation context, due to the high degree of context specificity of Arid1a function in regulating cell identity and plasticity.

# Materials and methods

**Key resources table**

| Reagent type (species) or resource | Designation | Source or reference | Identifiers | Additional information |
|---|---|---|---|---|
| Strain, strain background (*M. musculus*) | KC-RIK-shArid1a.6421 | This paper | | |
| Strain, strain background (*M. musculus*) | KC-RIK-shArid1a.1803 | This paper | | |
| Strain, strain background (*M. musculus*) | KC-RIK-shRen.713 | This paper | | |

*Continued on next page*

*Continued*

| Reagent type (species) or resource | Designation | Source or reference | Identifiers | Additional information |
|---|---|---|---|---|
| Antibody | GFP antibody (chicken polyclonal) | Abcam | Abcam Cat# ab13970, RRID:AB_300798 | |
| Antibody | GFP antibody (rabbit monoclonal) | Cell Signaling Technology | Cell Signaling Technology Cat# 2956P, RRID:AB_10828931 | |
| Antibody | mKate2 (rabbit polyclonal) | Evrogen | Evrogen Cat# AB233, RRID:AB_2571743 | |
| Antibody | Arid1a (rabbit polyclonal) | Cell Signaling Technology | Cell Signaling Technology Cat# 12354, RRID:AB_2637010 | |
| Antibody | Arid1a (mouse monoclonal) | Santa Cruz | Santa Cruz Biotechnology Cat# sc-32761, RRID:AB_673396 | |
| Antibody | Ki67 (mouse) | BD Biosciences | BD55609 | |
| Antibody | Cpa1 (goat polyclonal) | R and D | R and D Systems Cat# AF2765, RRID:AB_2085841 | |
| Antibody | Clusterin (goat polyclonal) | Santa Cruz | Santa Cruz Biotechnology Cat# sc-6419, RRID:AB_673567 | |
| Antibody | Sox9 (rabbit polyclonal) | Millipore | Millipore Cat# AB5535, RRID:AB_2239761 | |
| Antibody | CD19 Troma III (rat monoclonal) | Developmental Studies Hybridoma Bank | DSHB Cat# TROMA-III, RRID:AB_2133570 | |
| Antibody | Cleaved casp3 (rabbit monoclonal) | Cell Signaling Technology | Cell Signaling Technology Cat# 9664, RRID:AB_2070042 | |
| Antibody | CK8 (mouse monoclonal) | Biolegend | BioLegend Cat# 904801, RRID:AB_2565043 | |
| Antibody | CD45 (rat monoclonal) | Abcam | Abcam Cat# ab25386, RRID:AB_470499 | |
| Antibody | E cadherin (monoclonal) | BD Biosciences | BD Biosciences Cat# 610181, RRID:AB_397580 | |
| Antibody | donkey- anti-chicken CF488 | Sigma | Sigma-Aldrich Cat# SAB4600031, RRID:AB_2721061 | |
| Antibody | goat-anti-chicken AF488 | Life Technologies | Molecular Probes Cat# A-11039, RRID:AB_142924 | |
| Antibody | goat anti-rabbit AF594 | ThermoFisher | Thermo Fisher Scientific Cat# A-11037, RRID:AB_2534095 | |
| | Nextera Tn5 transposase | Illumina | FC-121–1030 | |
| Commercial assay or kit | Taqman copy number assay for GFP | Life Technologies/Thermo Fisher | 4400291 | |
| | NEBNext High-Fidelity 2x PCR Master Mix | New England Biolabs | NEB M0541 | |
| Software, algorithm | MSigDB database | http://software.broadinstitute.org/gsea/msigdb | | |
| Software, algorithm | BEDTools suite | http://bedtools.readthedocs.io | | |
| Software, algorithm | Homer v4.5 | http://homer.ucsd.edu/ | | |

*Continued on next page*

*Continued*

| Reagent type (species) or resource | Designation | Source or reference | Identifiers | Additional information |
|---|---|---|---|---|
| Other | ImmPress HRP | Vector | Vector Laboratories Cat# MP-7451, RRID:AB_2631198 | |
| Other | SuperSignal West Femto substrate | ThermoFisher | | |
| Other | Doxycycline diet (625 mg/kg) | Harlan | | |
| Other | M15 + LIF media | PMID: 24395249 | | |
| Other | DNase I | Sigma | DN25-100MG | |
| Other | Soybean Trypsin Inhibitor | Sigma | T9003-250mg | |
| Other | Dispase | Roche | | |

## Generation and authentication of ES cell clones

KC-RIK ES cells were cultured in M15 +LIF media and targeted as described previously (*Dow et al., 2012*; *Saborowski et al., 2014*). Clones were selected with hygromycin (Roche) and subjected to functional testing and copy number validation. For functional testing, clones were treated cultured ±Cre expressing Adenovirus (University of Iowa) cultured for 3 days ± doxycycline. Clones that were GFP +as assessed by flow cytometry in the Cre +Dox condition were assessed for single integration into the CHC locus using the Taqman copy number assay for GFP (Life Technologies) on a ViiA7 RT–PCR machine (Life Technologies). Positive clones were expanded and switched to KOSR + 2I 2 days prior to blastocyst injection. The identity and genotype of the ES was authenticated by allele-specific PCRs, and normal genomic status was confirmed by high-resolution Comparative Genomic Hybridization (CGH) array, as previously described (*Saborowski et al., 2014*). ES were confirmed to be negative for mycoplasma and other microorganisms before injection.

## Mice

All animal experiments in this study were performed in accordance with a protocol approved by the Memorial Sloan-Kettering Institutional Animal Care and Use Committee. Experimental animals were maintained on a mixed strain background. Doxycycline diet (625 mg/kg, Harlan) was changed twice weekly. Chimeric mouse cohorts were generated through the Mouse Genetics core and Center for Pancreatic Cancer Research (CPCR) at MSKCC. For caerulein experiments, mice were given eight hourly intraperitoneal injections of PBS or 50 µg/kg caerulein one day apart. shRNA and genotyping primer sequences shRNAs targeting the following sequences were cloned into mir30-based targeting constructs as described previously (*Dow et al., 2012*). shRen: CAGGAATTATAATGCTTATCTA shArid1a.6421: AGCCTGGAGAAGTTGTATAGTA shArid1a.1803: CCAGGAGCTTTCTCAAGATTCA

Genotyping of chimeric mice and their progeny was performed as previously described using a common *Col1a1* primer paired with a transgene specific primer:

Common: TTCAGACAGTGACTCTTCTGC shRen.713: GTATAGATAAGCATTATAATTCCTA shArid1a.6421: TATACTATACAACTTCTCCAGGCG shArid1a.1803: TATGAATCTTGAGAAAGCTCCTGT

## Immunofluorescence and immunohistochemistry

Tissues were fixed overnight in 4% paraformaldehyde prior to paraffin embedding. Five-micron sections were rehydrated with an alcohol series and subjected to antigen retrieval with citrate buffer or Tris pH 9.0. Slides were blocked in PBS with 5% BSA and 0.05% Triton-X. Primary antibody staining was performed overnight at 4°C or for 1 hr at room temperature. The following primary antibodies were used: GFP (Abcam Cat# ab13970, RRID:AB_300798 and Cell Signaling Technology Cat# 2956S, RRID:AB_1196615), mKate2 (Evrogen Cat# AB233, RRID:AB_2571743), Arid1a (Cell Signaling Technology Cat# 12354, RRID:AB_2637010 and SCBT sc-32761, RRID:AB_673396), Ki67 (BD BD55609), Cpa1 (R and D AF2765), Clusterin (SCBT sc-6419, RRID:AB_673567), Sox9 (Millipore AB553 , RRID:AB_2239761), CK19 Troma III (Developmental Studies Hybridoma Bank AB_2133570), Cleaved-Caspase 3 (Cell Signaling Technology Cat# 9664, RRID:AB_2070042), CK8 (BioLegend Cat#

904801, RRID:AB_2565043), E-cadherin (BD BD610181). For immunofluorescence, the following secondary antibodies were used: donkey-anti-chicken CF488 (Sigma-Aldrich Cat# SAB4600031, RRID: AB_2721061), goat-anti-chicken AF488 and AF647 (Molecular Probes Cat# A-11039, RRID:AB_142924), donkey anti-rabbit AF594 (Molecular Probes Cat# A-21207, RRID:AB_141637), goat anti-rabbit AF594 (Life Technologies A11037). Slides were counterstained with DAPI and mounted in Pro-Long Gold (Life Technologies). For Arid1a, pERK and Mesothelin immunohistochemistry, Vector ImmPress HRP kits were used for secondary detection. Tissues were counterstained with haematoxylin or Alcian blue, dehydrated and mounted with Permount (Fisher). Images were acquired on a Zeiss AxioImager microscope using Axiovision software. CD45 Immunohistochemistry was performed on a Bond Rx autostainer (Leica Biosystems) with heat mediated antigen retrieval using citrate buffer. Antibodies used were rat monoclonal CD45 primary antibody (Abcam 25386). Rabbit anti rat (Vector Laboratories, AI 4001) was used as secondary and Bond Polymer Refine Detection (Leica Biosystems) was used according to manufacturer's protocol. Sections were then counterstained with hematoxycilin, dehydrated and film coverslipped using a TissueTek-Prisma and Coverslipper (Sakura). Whole slide scanning (40x) was performed on an Aperio AT2 (Leica Biosystems). Quantification of GFP, Cpa1, Ki67 and c-Casp3 was performed using FIJI on sections from >3 animals per condition. For histological analysis of mucinous lesions, PanINs were classified and graded by a veterinary pathologist blinded to genotype, using established criteria (Gopinathan et al., 2015). IPMNs were scored based on lesion size, epithelial lining cells (squamous or low cuboidal with ductal traits) and stroma. For assessing inflammation in the caerulein-induced regeneration experiment, samples were scored 0–4, based on edema, intensity of inflammatory cells in the lesions and inflammation in the interstitial space. Statistical analyses were performed using unpaired t-test in Prism 7. Graphs displayed as mean ±SD.

## Western blot

Pancreas tissue was snap frozen in liquid nitrogen was lysed on ice with RIPA buffer containing HALT Protease and Phosphatase inhibitor cocktail. Primary antibody incubation was performed overnight at 4°C in Tris-buffered saline containing 5% milk and 0.05% Tween-20. The following antibodies were used: GFP (Cell Signaling Technology 2956), Arid1a (Cell Signaling Technology 12354), Cpa1 (R and D AF2765), Amylase (Sigma A8273) and Actin-HRP (Sigma A3854). Blots were developed with SuperSignal West Femto substrate (ThermoFisher).

## Pancreas epithelial cell isolation

Pancreata from KC-RIK-shRNA mice were chopped with scissors, finely minced with a blade and dissociated with digestion buffer: 1 mg/ml Collagenase V dissolved in HBSS with $Mg^{2+}$ and $Ca^{2+}$ supplemented with 0.1 mg/ml DNase I (Sigma, DN25-100MG) and 0.1 mg/ml Soybean Trypsin Inhibitor (Sigma T9003-250mg) for 30 min while rotating at 37°C. After washing with PBS, samples were further digested with 0.05% Trypsin-EDTA at 37°C. Trypsin digestion was neutralized with FACS buffer (10 mM EGTA and 2% FBS in $Ca^{2+}$ $Mg^{2+}$in PBS) containing STI. After an additional wash, samples were subsequently digested with 2 U/ml dispase (Roche) in PBS containing DNase I and STI for 20 min at 37°C. Samples were then washed in FACS buffer containing DNase I and STI, filtered through a 40 µm strainer and subjected to FACS. Samples were collected in FACS buffer with an additional 10% FBS for ATAC-seq or sorted directly into TRIzolLS for RNAseq. Each sample was used for either RNAseq or ATAC-seq.

## RNAseq analysis

Total RNA was isolated from cells sorted into TRIzolLS and assessed using a BioAnalyzer (Agilent). Sequencing and library preparation was performed at the Integrated Genomics Operation (IGO) at MSKCC. Ribosomal RNA was depleted using RiboZero (Illumina) and stranded KAPA RNA libraries were prepared from 1 ug RNA. Approximately 40 million paired-end 50 bp reads were sequenced per replicate on a HiSeq 2000. Resulting RNA-Seq data was analyzed by removing adaptor sequences using Trimmomatic (Bolger et al., 2014). Reads were then aligned to the mouse genome (NCBI37/mm9) using STAR aligner (Dobin et al., 2013) and transcript counting was performed by HTseq (Anders et al., 2015). Differential expression analysis was performed using the Limma and Voom R/Bioconductor packages with an FDR cutoff of 0.05 (Law et al., 2014; Ritchie et al., 2015).

RNAseq data have been deposited to GEO under series GSE114576. Pathway analysis was performed using the GSEA pre-ranked mode against signatures in the MSigDB database (http://software.broadinstitute.org/gsea/msigdb) and published expression signatures in organoid and mouse models (*Boj et al., 2015*; *Holmstrom et al., 2011*; *Jiang et al., 2016*; *Morris et al., 2014*).

## Quantitative Real-Time polymerase chain reaction (qRT-PCR) analysis

Total RNA was isolated using the TRIZOL (Thermo Fisher Scientific), and cDNA was obtained from 500 ng of RNA using the Transcriptor First Strand cDNA Synthesis Kit (Roche) after treatment with DNAse I (Invitrogen) and according to manufacturer's instructions. The following gene-specific primer sets for mouse sequences were used: Arid1a_F TCCCAGCAAACTGCCTATTC, Arid1a_R CATATCTTCTTGCCCTCCCTTAC, Agr2_F ACAACTGACAAGCACCTTTCTC, Agr2_R GTTTGAGTATCGTCCAGTGATGT, Klk1_F GCCCTGGCAAGTGGCTGTGT, Klk1_R AAGCCGGTGTTGGGCAGAGG, Msl1_F CCCATCGAAGTGGTCAGTCTC, Msln_R GGTGTATGACGGTCAGCTTAGA, Cel_F AAGTTGCCCGTGAAAAAGCAG, Cel_R ATGGTAGCAAATAGGTGGCCG, Amy1_F TCACACGGGTGATGTCAAGTT, Amy1_R GTCTGGGTTAATGCTCACTTCTT, Hprt_F TCAGTCAACGGGGGACATAAA and Hprt_R GGGGCTGTACTGCTTAACCAG. qRT-PCR was carried out in triplicate (5 cDNA ng/reaction) using SYBR Green PCR Master Mix (Applied Biosystems) on the ViiA 7 Real-Time PCR System (Life technologies). Hprt served as endogenous normalization control.

## ATACseq analysis

50,000 mKate2$^+$ cells isolated by FACS were lysed in 50 ul lysis buffer (*Buenrostro et al., 2015*) and subjected to transposition with Nextera Tn5 transposase according to manufacturer's instructions (Illumina FC-121–1030). DNA was eluted from a MinElute column in 11.5 ul elution buffer (Qiagen). ATAC libraries were constructed using the NEBNext High-Fidelity 2x PCR Master Mix (NEB M0541) as in Buenrostro et al with the following modifications: DNA was PCR-amplified for 1 cycle of 5' at 72°C and 30' at 98°C followed by 12 cycles of 10' at 98°C, 30' at 63°C and 1' at 72°C. Amplified DNA was purified on a Qiagen MinElute column and eluted in 22 ul of Qiagen elution buffer. Purified libraries were assessed using a Bioanalyzer High-Sensitivity DNA Analysis kit (Agilent). Paired-end 50 bp reads were sequenced with the Center for Epigenetics Research at Memorial Sloan Kettering Cancer Center. Reads were trimmed for quality and Illumina adapter sequences using 'trim_galore' then aligned to mouse genome assembly mm9 with bowtie2 using the default parameters. Aligned reads with the same start site and orientation were removed using the Picard tool MarkDuplicates (http://broadinstitute.github.io/picard/). Density profiles for genome browser tracks were created using the BEDTools suite (http://bedtools.readthedocs.io). Enriched regions were discovered using MACS2 and scored against input sequence (fold change >2 and p-value<0.001). Dynamic regions between two conditions were discovered using a similar method, with the second ATAC library replacing input. Peaks were then filtered against genomic 'blacklisted' regions (http://mitra.stanford.edu/kundaje/akundaje/release/blacklists/mm9-mouse/mm9-blacklist.bed.gz) and those within 500 bp were merged. All genome browser tracks and read density tables were normalized to a sequencing depth of ten million mapped reads. K-means clustering was performed on all dynamic peaks, with the optimal k determined by silhouette analysis. Dynamic peaks were annotated using linear genomic distance and motif signatures were obtained using the 'de novo' approach with Homer v4.5 (http://homer.ucsd.edu/). For analyses of TF density, transcription factor ChIP-seq enriched regions were discovered by comparison with matched input using MACS2, as with the ATAC-seq data. TF peaks overlapping sites decreasing in accessibility were classified using 'bedtools intersect' with an overlap of 1 bp. Gene density of these peaks was calculated using RefSeq annotated gene boundaries ± 30 kb of gene start or end, and these gene names were then matched with either differentially expressed genes, or genes with expression not changing by more than 10%. ATACseq data have been deposited to GEO under accession GSE114567.

## Acknowledgements

We would like to thank Zhen Zhao and So-Young Kim (MSKCC Center for Pancreatic Cancer Research) for assistance with ESC-GEMM production, Janelle Simon, Sha Tian and Sarah Ackermann for assistance with mouse colonies, and the other Lowe laboratory members for helpful advice and discussions. We would also like to thank Rohit Chandwani for assistance with ATACseq, and

Meredith E Pittman for advice on pathology analysis. We would like to thank Yu-Jui Ho for assistance with RNAseq analysis. We would additionally like to thank Harini Babu (Histowiz) for assistance with CD45 immunohistochemistry. This work was supported by the Lustgarten Foundation; GL was supported by NIH F32 grant 1F32CA177072-01 and American Cancer Society fellowship ACS PF-13-037-01-DMC; DAC was recipient of a postdoctoral fellowship from the Spanish Fundación Ramón Areces; SWL is the Geoffrey Beene chair for Cancer Biology and a Howard Hughes Medical Institute investigator.

## Additional information

### Funding

| Funder | Grant reference number | Author |
| --- | --- | --- |
| Lustgarten Foundation | 388171 | Geulah Livshits<br>Direna Alonso-Curbelo<br>John P Morris IV |
| National Cancer Institute | 1F32CA177072-01 | Geulah Livshits |
| American Cancer Society | PF-13-037-01-DMC | Geulah Livshits |
| Lustgarten Foundation | Research Investigator Award | Scott W Lowe |
| National Cancer Institute | CA13106 | Scott W Lowe |
| National Cancer Institute | Cancer Center Core Grants P30-CA008748 | Scott W Lowe |

The funders had no role in study design, data collection and interpretation, or the decision to submit the work for publication.

### Author contributions

Geulah Livshits, Conceptualization, Formal analysis, Funding acquisition, Investigation, Visualization, Writing—original draft, Writing—review and editing; Direna Alonso-Curbelo, Data curation, Formal analysis, Validation, Investigation, Methodology, Writing—original draft, Writing—review and editing; John P Morris IV, Resources, Formal analysis, Validation, Methodology, Writing—review and editing; Richard Koche, Software, Formal analysis, Validation, Investigation; Michael Saborowski, Resources, Methodology; John Erby Wilkinson, Formal analysis, Validation, Investigation; Scott W Lowe, Conceptualization, Resources, Supervision, Methodology, Project administration, Writing—review and editing

### Author ORCIDs

Geulah Livshits http://orcid.org/0000-0002-3835-7377
Direna Alonso-Curbelo http://orcid.org/0000-0001-6674-3059
Richard Koche http://orcid.org/0000-0002-6820-5083
Scott W Lowe http://orcid.org/0000-0002-5284-9650

### Ethics

Animal experimentation: This study was performed in accordance with the guidelines set by the National Institutes of Health in the Guide for the Care and Use of Laboratory Animals. Animals were handled according Institutional Animal Care and Use committees (IACUC) protocols #11-06-016 and 11-06-018 at MSKCC. Animals were monitored for signs of ill-health by veterinary staff at the Research Animal Resource Center (RARC) at MSKCC and efforts were made to minimize suffering.

### Decision letter and Author response
Decision letter https://doi.org/10.7554/eLife.35216.037
Author response https://doi.org/10.7554/eLife.35216.038

# Additional files

## Supplementary files
• Transparent reporting form
DOI: https://doi.org/10.7554/eLife.35216.023

## Data availability
Sequencing data have been deposited in GEO under accession numbers GSE114576 and GSE114567.

The following datasets were generated:

| Author(s) | Year | Dataset title | Dataset URL | Database, license, and accessibility information |
| --- | --- | --- | --- | --- |
| Lowe SW | 2018 | RNAseq data | http://www.ncbi.nlm.nih.gov/geo/query/acc.cgi?acc=GSE114576 | Publicly available at the NCBI Gene Expression Omnibus (accession no. GSE114576) |
| Lowe SW | 2018 | ATACseq data | http://www.ncbi.nlm.nih.gov/geo/query/acc.cgi?acc=GSE114567 | Publicly available at the NCBI Gene Expression Omnibus (accession no. GSE114567) |

The following previously published datasets were used:

| Author(s) | Year | Dataset title | Dataset URL | Database, license, and accessibility information |
| --- | --- | --- | --- | --- |
| Boj SF, Hwang C-I, Baker LA, Chio IIC, Engle DD, Corbo V, Jager M, Ponz-Sarvise M, Tiriac H, Spector MS | 2015 | Expression Analysis of Normal and Neoplastic Mouse Pancreatic Ductal Organoids | https://www.ncbi.nlm.nih.gov/geo/query/acc.cgi?acc=GSE63348 | Publicly available at the NCBI Gene Expression Omnibus (accession no: GSE63348) |
| Krah NM, De La O J-P, Swift GH, Hoang CQ, Willet SG, Chen Pan F, Cash GM, Bronner MP, Wright CV, MacDonald RJ | 2015 | Effects on the transcriptome of adult mouse pancreas (principally acinar cells) by the inactivation of the Ptf1a gene in vivo | https://www.ncbi.nlm.nih.gov/geo/query/acc.cgi?acc=GSE70542 | Publicly available at the NCBI Gene Expression Omnibus (accession no: GSE70542) |
| Hiraoka N, Yamazaki-Itoh R, Ino Y, Mizuguchi Y, Yamada T, Hirohashi S, Kanai Y | 2011 | Multistep pancreatic carcinogenesis: epithelial cells | https://www.ncbi.nlm.nih.gov/sites/GDSbrowser?acc=GDS3836 | Publicly available at the NCBI GDSbrowser (accession no: GDS3836) |
| Jiang M, Azevedo-Pouly A, Deering TG, Hoang CQ, DiRenzo D, Hess DA, Konieczny SF, Swift GH, MacDonald RJ | 2016 | MIST1 and PTF1 Collaborate in Feed-forward Regulatory Loops that Maintain the Pancreatic Acinar Phenotype in Adult Mice | https://www.ncbi.nlm.nih.gov/geo/query/acc.cgi?acc=GSE86290 | Publicly available at the NCBI Gene Expression Omnibus (accession no: GSE86290) |

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
