## [Decision Letter]

Thank you for submitting your article "Arid1a restrains Kras-dependent changes in acinar cell identity" for consideration by *eLife*. Your article has been reviewed by three peer reviewers, and the evaluation has been overseen by a Reviewing Editor and Fiona Watt as the Senior Editor. The reviewers have opted to remain anonymous.

The reviewers have discussed the reviews with one another and the Reviewing Editor has drafted this decision to help you prepare a revised submission.

Summary:

In their manuscript, Livshits et al. investigate the role of the chromatin remodeling protein Arid1a in pancreatic carcinogenesis and acinar cell identity. To study the consequences of Arid1a-deficieny on acinar fate in the presence and absence of oncogenic Kras activation the authors utilize a very elegant ES cell RNAi based mouse model that facilitates inducible and reversible targeting of Arid1a independent on the Cre-mediated activation of oncogenic Kras during embryogenesis. In addition to phenotypic analyses using the described mouse model system the manuscript provides first mechanistic insights into pancreatic carcinogenesis in the context of Arid1a deficiency using RNA-Seq and ATAC-Seq analysis. Together, the authors demonstrate that Arid1a can function as a barrier to Kras-driven pancreatic cancer initiation, albeit the impact of Arid1a-deficiency underlies a significant context-dependency. The manuscript significantly extends our understanding of the involvement of chromatin remodeling processes in general and the function of Arid1A in particular in pancreatic cancer development. All together the manuscript is perfectly written and follows a logical order and the conclusions drawn from the data are largely justified.

1) Please provide more information about the lesions in the mice. The authors state that the ductal lesions found in the Arid1a KD mice can be described as PanIN or IPMN-lesions. In Figure 2D only PanINs have been quantified. How have the lesions been distinguished from IPMNs so that only PanINs were counted? Do the Arid1a KD mice give rise to both, PanINs and IPMN? As BRG1-deficiency in the context of oncogenic Kras activation in the pancreas results to development of IMPN lesions (v. Figura et al., 2014), formation of IPMN might be a likely phenotype in Arid1A kd mice. As the severity of PanINs can be more important than their total numbers, it is important to illustrate the PanIN quantification in Figure 2D for the different PanIN stages separately. Please state how many mice have been observed until an age of app. 1 year (Figure 2—figure supplement 2A)? Was there a significant number of mice to justify the authors´ statement that Arid1a kd animals do not progress to PDAC? Can the authors show that Arid1a is still absent after one year of Dox treatment? How exactly have those lesions been quantified? Counting the lesions per field might lead to misleading numbers, if the Arid1 kd pancreata suffer from pancreatic atrophy. How is the number of ADMs normalized to pancreatic size?

Figure 3J: Arid1a kd and control mice are challenged with caerulein to induce pancreatitis. The authors state that they observe comparable ADM formation and pancreatic inflammation in both genotypes. Could the authors please quantify the inflammatory response (e.g. by using a scoring system considering edema, infiltration of inflammatory cells into the pancreas etc.) to objective the statement of comparable inflammation? In the same line: is the caerulein-induced damage comparable in both models? If blood samples from the mice were available, lipase and amylase serum levels could be assessed. Comparable damage in both models is important to justify the statement of equal regenerative potential in both models.

2) Please provide more information about the transcriptional analysis. The unsupervised hierarchical clustering depicted in Figure 6B shows 3 samples for each Arid1a shRNA. In contrast, the PCA in Figure 6A only shows two dots for A1A.1803. Is the dot hidden behind another one or have the authors excluded one sample? If so, what has been the rationale for excluding the sample? In the same line: in ATAC-Seq, there are four replicates for shRen, but 3 in shArid1A. Why? Has a shArid1A sample been excluded? The transcriptional analysis in cell extracts from KC-Ren and KC-shArid1a cells reveals genes and signatures associated with carcinogenesis and acinar cell fate. Could the authors please validate the distinct expression of some of these genes in their isolated cells and in the tissue of the mice? Have RNA-Seq and ATAC-Seq been performed using cells from identical animals?

3) Please solidify the analysis of transcriptional programs and chromatin accessibility by demonstrating clearer relationships between the two. For example, the text states "gene sets comprising the targets of these transcription factors were enriched among the downregulated genes in GSEA analysis of transcriptional profiles derived from shArid1a animals (Figure 7H)." However, Figure 7H only shows that genes whose chromatin accessibility decreased showed reduced expression. The authors should analyze the data to correlate expression and accessibility at target sites of key acinar transcription factor binding sites, and separately determine the accessibility at genes whose expression changes in shArid1a.

---

## [Author Response]

[…] 1) Please provide more information about the lesions in the mice. The authors state that the ductal lesions found in the Arid1a KD mice can be described as PanIN or IPMN-lesions. In Figure 2D only PanINs have been quantified. How have the lesions been distinguished from IPMNs so that only PanINs were counted? Do the Arid1a KD mice give rise to both, PanINs and IPMN? As BRG1-deficiency in the context of oncogenic Kras activation in the pancreas results to development of IMPN lesions (v. Figura et al., 2014), formation of IPMN might be a likely phenotype in Arid1A kd mice. As the severity of PanINs can be more important than their total numbers, it is important to illustrate the PanIN quantification in Figure 2D for the different PanIN stages separately.

We now quantify and classify ductal lesions as ADM, PanIN (1, 2, or 3), or IPMN, confirming that Arid1a suppression in adult pancreatic cells following expression of mutant Kras during embryonic development predominantly drives development of low grade PanIN lesions (see new Figure 2D and subsection “Acute Arid1a knockdown induces rapid changes in adult acinar cell fate in the setting of Kras^G12D^”, second paragraph). Regarding PanIN grades, histological analysis 2 weeks after shArid1a inhibition in this adult cohort reveals the majority of lesions are consistent with PanIN-1, though a small number of PanIN-2 lesions were present as well. No PanIN-3 lesions were observed. Similarly, few IPMNs were identified. These data have been included in new Figure 2D. The differences between Arid1a KD and Brg1-deficiency mouse models are discussed in Results subsection “Temporal context dictates the outcome of Arid1a depletion” (first paragraph), and in the fourth and fifth paragraphs of the Discussion.

Please state how many mice have been observed until an age of app. 1 year (Figure 2—figure supplement 2A)? Was there a significant number of mice to justify the authors´ statement that Arid1a kd animals do not progress to PDAC? Can the authors show that Arid1a is still absent after one year of Dox treatment?

A substantial portion of the 66 KC-shArid1a mice placed on dox as adults became morbid and required euthanasia prior to 1 year of age in the absence of PDAC, likely due to the exocrine dysfunction as described in the text. 11 shArid1a mice were analyzed at >1 year of age; of these, one had progressed to cancer, while the rest retained mucinous lesions as pictured. We have confirmed the stability of Arid1a knockdown in mucinous PanIN lesions in this aged cohort (Figure 2—figure supplement 2B, 2C).This frequency of PDAC development after a year is consistent with historically reported rates in mouse models where mutant Kras is expressed in the developing pancreas in the absence of genetic modification of other tumor suppressors (subsection “Acute Arid1a knockdown induces rapid changes in adult acinar cell fate in the setting of Kras^G12D^”, last paragraph). This supports a role for Arid1a suppression constraining Kras driven preneoplastic development rather than malignant progression as addressed in the original and revised manuscript.

How exactly have those lesions been quantified?

Quantification was performed by a veterinary pathologist experienced in murine pancreas histology (J. Wilkinson) blinded to genotype. To clarify our methods to the reader, we added a supplementary figure (Figure 2—figure supplement 1G) showing our scoring criteria and examples of the quantified lesions (ADM, PanIN, and IMPN), and extended our Materials and method’s description.

Counting the lesions per field might lead to misleading numbers, if the Arid1 kd pancreata suffer from pancreatic atrophy. How is the number of ADMs normalized to pancreatic size?

While we agree that counting lesions per field can be misleading in the setting of pancreatic atrophy, shArid1a mice displayed a slightly *increased* pancreas:body weight ratio compared to controls at the time when mice were analyzed, likely due to the stromal expansion that accompanies preneoplastic initiation (von Figura et al.,2014) (see new Figure 2—figure supplement 1D, described in subsection “Acute Arid1a knockdown induces rapid changes in adult acinar cell fate in the setting of Kras^G12D^”, second paragraph). As suggested, we also updated the graphs that reported “lesion per field” to express lesions relative to mm2 of tissue area analyzed (new Figure 2D).

Figure 3J: Arid1a kd and control mice are challenged with caerulein to induce pancreatitis. The authors state that they observe comparable ADM formation and pancreatic inflammation in both genotypes. Could the authors please quantify the inflammatory response (e.g. by using a scoring system considering edema, infiltration of inflammatory cells into the pancreas etc.) to objective the statement of comparable inflammation? In the same line: is the caerulein-induced damage comparable in both models? If blood samples from the mice were available, lipase and amylase serum levels could be assessed. Comparable damage in both models is important to justify the statement of equal regenerative potential in both models.

To exclude the possibility that genotype specific differences in acute pancreatic damage in response to caerulein prevents accurate evaluation of the regenerative capacity of shArid1a expressing pancreatic cells, we have more fully characterized the time course of pancreatic inflammation, exocrine stress, and exocrine regeneration in shArid1a and shRen expressing mice. Scoring inflammation based on histological analysis and CD45 immunohistochemistry reveals similar levels of inflammation at D2 post caerulein in both shArid1a and shRen expressing mice, as well as comparable resolution at D9. (Figure 3—figure supplement 2A, B).While no blood samples were available to measure serum pancreatic enzyme levels as proxy for pancreatic damage, we compared molecular markers of acinar stress and acinar differentiation after caerulein treatment. As shown in new Figure 3—figure supplement 2C, caerulein treatment induced a comparable increase in clusterin, a well-established anti-apoptotic gene that is transiently expressed in damaged acini following pancreatic damage and downregulated upon regeneration (Siveke et al.,2008, Morris et al., 2010), in both genotypes at D2 that is markedly reduced upon regeneration (D9). Furthermore, we now show an inverse expression pattern of the differentiated acinar marker Cpa1 in both shRen and shArid1a acinar cells undergoing injury-induced regeneration (Figure 2—figure supplement 3A, B). Importantly, as shown in Figure 3—figure supplement 3B, Arid1a levels remained suppressed at all stages of response and regeneration following acute pancreatitis, thus confirming the absence of counter selection against Arid1a suppressed cells and expansion of Arid1a expressing cells. These results provide strong evidence that Arid1a suppression does not alter the damage:regeneration capacity of adult acinar cells and that Arid1a is dispensable for restoring acinar fate after injury. These issues are now detailed fully in the results (subsection “Arid1a is not required for acinar cell maintenance or regeneration in the normal pancreas”, last paragraph).

2) Please provide more information about the transcriptional analysis. The unsupervised hierarchical clustering depicted in Figure 6B shows 3 samples for each Arid1a shRNA. In contrast, the PCA in Figure 6A only shows two dots for A1A.1803. Is the dot hidden behind another one or have the authors excluded one sample? If so, what has been the rationale for excluding the sample?

There are 3 dots for A1A.1803 in the PCA plot in Figure 6A, with one inadvertently cropped during composition of the figure. We updated Figure 6A to correct this error. Importantly, the results are not affected by this correction.

In the same line: in ATAC-Seq, there are four replicates for shRen, but 3 in shArid1A. Why? Has a shArid1A sample been excluded?

No samples were excluded – we were only able to generate ATAC libraries of sufficient quality from 3 shArid1a and 4 control animals and all available results are presented.

The transcriptional analysis in cell extracts from KC-Ren and KC-shArid1a cells reveals genes and signatures associated with carcinogenesis and acinar cell fate. Could the authors please validate the distinct expression of some of these genes in their isolated cells and in the tissue of the mice?

We now include validation of the RNAseq data by qRT-PCR performed in RNA extracted from FACS-sorted shRNA expressing epithelial cells 5 days after dox administration in adult mice with histologically validated Arid1a knockdown (Figure 6—figure supplement 1A). qRT-PCR analyses confirmed significant deregulation of the expression of both carcinogenesis-associated genes (Agr2, Mesothelin) and acinar identity genes (Cel, Klk1) upon Arid1a knockdown (Figure 6E). Moreover, histological validation of select up- and down-regulated genes at 2 weeks post dox treatment shows that early transcriptional changes induced by Arid1a suppression (e.g. mesothelin upregulation, and amylase downregulation) are maintained in the resultant metaplastic pancreas (Figure 6—figure supplement 1B).

Have RNA-Seq and ATAC-Seq been performed using cells from identical animals?

Technical challenges involved in dissociating epithelial cells from fibrotic pancreata undergoing Kras driven metaplasia precluded our ability to collect sufficient cells to perform both ATAC and RNA SEQ from the same animals. This is clarified in the revised Materials and methods section.

3) Please solidify the analysis of transcriptional programs and chromatin accessibility by demonstrating clearer relationships between the two. For example, the text states "gene sets comprising the targets of these transcription factors were enriched among the downregulated genes in GSEA analysis of transcriptional profiles derived from shArid1a animals (Figure 7H)." However, Figure 7H only shows that genes whose chromatin accessibility decreased showed reduced expression. The authors should analyze the data to correlate expression and accessibility at target sites of key acinar transcription factor binding sites, and separately determine the accessibility at genes whose expression changes in shArid1a.

We performed a comprehensive analysis of gene expression and chromatin accessibility at targets of pancreatic TFs necessary for maintenance of acinar differentiation (Nr5a2, Ptf1a) and acinar regeneration (Pdx1) that previously demonstrated to constrain Kras driven preneoplastic development (von Figura et al.,2014, Krah et al.,2015, Roy et al.,2016). We compared ATAC-SEQ profiles following Arid1a suppression to high confidence, experimentally-defined datasets of TF binding sites defined by ChIP-seq (Holmstrom et al.,2011, Hoang et al.,2016, Roy et al.,2016). This analysis showed a significant enrichment of TF binding sites at ATAC-seq peaks that lose accessibility upon Arid1a KD compared with sites not changing in accessibility (Nr5a2 and Ptf1a: p < 2.2e-16, Pdx1: p < 0.0016; Fisher's exact test). Composite plots of average ATAC-seq signal at these TF binding sites also demonstrated decreased accessibility in Arid1a KD compared with control (Figure 7E). Importantly, increased chromatin closing observed upon Arid1a KD at these TF target sites is associated with reduced gene expression at target genes, now documented in two ways. First, by associating these TF-ATAC sites with genes by linear genomic distance and then using GSEA, we were able to find significant enrichment of these peaks in genes that become downregulated upon Arid1a KD as compared with control (Figure 7F). Second, by computing the density of these TF-ATAC sites per gene, we found increased density in genes downregulated compared with genes whose expression is upregulated or unchanged (Figure 7H; mean TF density in DEG down: 1.53, DEG up: 0.72, p = 2.55e-14, Welch’s t test). While the TF density was calculated per gene, we note that this is not simply an effect of underlying gene size discrepancies in the different expression categories (median gene size is 24 kb and 21 kb in downregulated and upregulated sets, respectively). Lastly, we note that various intersections of ATAC-seq dynamics and TF binding sites failed to yield a correlation with genes that become upregulated upon Arid1a KD.